# Silica-associated proteins from hexactinellid sponges support an alternative evolutionary scenario for biomineralization in Porifera

Katsuhiko Shimizu[1] ✉, Michika Nishi[2], Yuto Sakate[2], Haruka Kawanami[3], Tomohiro Bito[3], Jiro Arima[3], Laia Leria [4] & Manuel Maldonado [4] ✉

Metazoans use silicon traces but rarely develop extensive silica skeletons, except for the early-diverging lineage of sponges. The mechanisms underlying metazoan silicification remain incompletely understood, despite significant biotechnological and evolutionary implications. Here, the characterization of two proteins identified from hexactinellid sponge silica, hexaxilin and perisilin, supports that the three classes of siliceous sponges (Hexactinellida, Demospongiae, and Homoscleromorpha) use independent protein machineries to build their skeletons, which become non-homologous structures. Hexaxilin forms the axial filament to intracellularly pattern the main symmetry of the skeletal parts, while perisilin appears to operate in their thickening, guiding extracellular deposition of peripheral silica, as does glassin, a previously characterized hexactinellid silicifying protein. Distant hexaxilin homologs occur in some bilaterians with siliceous parts, suggesting putative conserved silicifying activity along metazoan evolution. The findings also support that ancestral Porifera were non-skeletonized, acquiring silica skeletons only after diverging into major classes, what reconciles molecular-clock dating and the fossil record.

Sponges (phylum Porifera), being early-diverging metazoans, have the potential to inform about the first evolutionary steps of many metazoan traits, including biomineralization systems. Extant Porifera consist of four taxonomic classes, one characterized by calcareous skeletons (Calcarea) and three by siliceous ones (Demosponges, Hexactinellida, Homoscleromorpha). Understanding the intricate cellular and molecular mechanisms through which silicifying cells (sclerocytes) of sponges polymerize silicon dissolved in seawater (silicic acid= dSi) to build their skeletal parts of silica (spicules) is of enormous evolutionary interest, but also rises important biotechnological expectations related to fiber optic developments[1–3], architectural materials and solutions[4–6], and mammalian bone

regeneration[7–9], among others[10] (Supplementary Note 1). For decades, it was a mystery why vertebrates require trace amounts of silicon in their diet for correct bone formation[11] and why an enhanced silicon provision facilitates the regeneration of their bone lesions[12,13]. The discovery in 2015 that silicon enters into human osteoblasts through aquaglyceroporin (3, 7 and 9) channels located at their plasmalemma[14] provided a first insight. The finding somehow connects mammalian bone formation to sponge silicification, as sponges use related aquaglyceroporin channels to uptake dSi[15]. Other non-poriferan metazoans, apart from vertebrates, also use trace amounts of silicon. For instance, copepods of the genus *Calanus*, limpets of the genus *Patella*[16,17], and nudibranchs of the genus *Felimare*[18] are

[1]Platform for Community-based Research and Education, Tottori University, 4-101, Koyama-cho, Minami, Tottori 680-8550, Japan. [2]Division of Agricultural Science, Graduate studies of Sustainability Science, Tottori University Graduate School, 4-101, Koyama-cho, Minami, Tottori 680-8553, Japan. [3]Department of Life Environmental Agriculture, Faculty of Agriculture, Tottori University, 4-101, Koyama-cho, Minami, Tottori 680-8553, Japan. [4]Sponge Ecobiology and Biotechnology Group, Center for Advanced Studies of Blanes (CEAB, CSIC), Blanes 17300, Spain. ✉e-mail: kshimizu@tottori-u.ac.jp; maldonado@ceab.csic.es

known to have partially silicified teeth, organosiliceous shells occur in some brachiopod larvae[19], and some polychaetes have been suggested to have silicified chaetae[20,21]. In remarkable concomitance with the occurrence of partially silicified structures in those bilaterians, silicon transporters of several types have also been identified in the genome of *Calanus* copepods and *Lottia* limpets, *Lingula* brachiopods, and *Capitella* polychaetes[21]. The cell and molecular machineries involved in the polycondensation of the transported dSi to produce silica in these non-poriferan metazoans remain even less explored and understood than those of sponges, not to mention their evolutionary interconnections, if any. Thus, a deeper knowledge of the biological production of silica by sponges may not only unlock new routes for the controlled synthesis of silicas with lower cost and environmental impact than current industrial processes−an old biotechnological aspiration[22]−but may also help to assemble the evolutionary puzzle of biomineralization in metazoans.

A first milestone in the molecular understanding of sponge silicification was established by the discovery of the enzyme silicatein, a homolog of cathepsin L, in Demospongiae in 1998[23]. This protein forms a strand (axial filament) within the silica deposition vesicle (SDV) of the sclerocyte, initiating spicule formation by inducing deposition of concentric silica layers around the filament[24,25]. The identification of the silicatein sequences in different demosponge lineages sparked a wave of application studies aiming technological production of biogenic silica that continues to rise today[10]. In 2008, silicatein was reported from two hexactinellid sponges[26,27], apparently strengthening the traditional assumption that the molecular bases of silicification were common to all classes of siliceous sponges. However, those initial claims of silicatein in Hexactinellida are now considered to be either contamination or misinterpreted cathepsin-L enzymes that are involved in lysosomal digestion rather than silicification. This new interpretation has arisen from the accumulation of transcriptomic and genomic studies concurring that functional silicateins are absent from both Homoscleromorpha and Hexactinellida[15,28–32].

A second milestone in the molecular understanding of sponge silicification was established in 2015 by the discovery of glassin in the soluble fraction of silica extracts from two species of the hexactinellid genus *Euplectella*[33]. Surprisingly, glassin was unrelated to silicatein and any other sequenced protein. This finding revealed that hexactinellids have at least a silicifying protein of their own[34], but left unsolved how the silicifying machinery of the three siliceous classes relates to each other and to that of other metazoans, if any. The solvable nature of glassin also made it difficult to explain how such a protein could constitute the solid rod (i.e. the axial filament) that is observed within hexactinellid spicules and that is assumed to determine spicule shape during silicification.

Additional organic components have been claimed to be associated with the silica of hexactinellid sponges, such as chitin[35] and actin[36]. Although their exact functions remain under debate (Supplementary Note 2), there is general agreement that they do not act directly in the polymerization of the silicic acid, that is, they are not silica-polycondensing molecules.

The complex puzzle of the silicifying sponge proteins is partially clarified here by describing two additional proteins unrelated to glassin and silicatein, which were extracted from the silica of the hexactinellid sponges *Euplectella curvistellata* and *Vazella pourtalesii* (Fig. 1a−c). Through immunodetection, distinctive locations within the silica are revealed for the two new proteins and for glassin. The phylogenetic relationships and diversification patterns of these proteins are also analyzed. Protein expressions are examined as well, using transcriptomes of individuals of *V. pourtalesii* whose silicification was experimentally stimulated in a previous study[15]. Collectively, the findings improve the current understanding about how hexactinellid spicules are made, establish alternative molecular scenarios for the origin and evolution of silicification in Porifera, and suggest a potential connection to the "residual" silicification shown by some bilaterians.

## Results and discussion

### Identification and location of proteins in the silica

A polyacrylamide gel electrophoresis analysis of the protein constituents in the insoluble fraction of spicule extracts obtained from the hexactinellids *E. curvistellata* and *V. pourtalesii* revealed a 38 kDa protein band in both species (Fig. 2; Supplementary Fig. 1). There was also a second band, which was 23 kDa in *E. curvistellata* and 32 kDa in *V. pourtalesii*. An amino-terminal analysis of the 38 kDa protein of *E. curvistellata* revealed the FPAPDGNLHIYAIPVG oligopeptide (Fig. 2a, b), which was further identified in the genome of *E. curvistellata*[30,33] as being part of an unannotated gene that encodes a 359 amino acid (aa) protein, including a possible 34-aa propeptide at the N-terminus (Fig. 2b). The sequenced oligopeptide for the 23 kDa protein was identical to one known from glassin (GLSIN)[33]. We raised an antibody against peptide sequences derived from the 38 kDa protein. Western blots using this antibody revealed a signal at 38 kDa consistent with the monomeric form of this protein and a higher MW band of c. 70 kDa consistent with a dimeric form. We performed immunodetections which revealed a signal exclusively in the axial filament of the spicules (Fig. 2f, g, Supplementary Fig. 2), presumably because the protein is tagged by the antibody when it becomes exposed through irregular fractures of the silica, but not when it remained entirely encased in silica. While the Western blots show that the antibody can recognize a protein of the correct molecular weight, we never resolved the exact identity of the putative dimeric band that was also labeled by the antibody (Supplementary Fig. 1b). Therefore, it cannot be ruled out that the antibody could also bind an additional unidentified protein of similar molecular weight to that expected for dimers of the target protein. Since the target protein appears to be located in the hexactinellid axial filament, we named it "hexaxilin" (HXX). Unlike hexaxilin, glassin formed concentric rings, intercalated between the layers of the peripheral silica (Fig. 2h−j). Labeling of spicule cross-sections corroborated that glassin never contributes to the axial filament (Fig. 2j).

The N-terminus analysis of the 38 kDa protein of *V. pourtalesii* revealed a FPKSDGNLHI oligopeptide (Fig. 2c, d), further identified in the transcriptome[15] as being part of a gene related to the *hexaxilin* of *E. curvistellata*. It encodes a 358-aa protein (Fig. 2d), including a possible 34-aa pro-peptide at the N-terminus. In contrast, the 32 kDa protein had the N-terminus oligopeptide GLTQQQKRQI, which was further identified in three unannotated isoforms (α, β, γ; 84.2% to 92.4% identity) of the same gene. Each encoded a 265 aa sequence (Fig. 2e), consisting of a 20 aa putative propeptide followed by a Cys-rich domain (12 Cys) of 186 aa, and with a 59 aa-long, His-rich (14-15 His) sequence at the C-terminus. Immunodetection revealed that a signal presumably representing hexaxilin of *V. pourtalesii* occurs exclusively in the axial filament of spicules (Fig. 2m, n, Supplementary Fig. 2). In contrast, an antibody raised against the 32 kDa protein occurred in concentric rings, intercalated between the peripheral silica layers (Fig. 2o−q), following a pattern similar to that of glassin in *E. curvistellata*. As above, while the Western blots showed that the antibody can recognize a protein of the correct molecular weight, we never resolved the exact nature of additional bands that were also bound by the antibody and that we assumed to be perisilin dimers and trimers (Supplementary Fig. 1f). Therefore, it cannot be ruled out that the antibody may also bind unidentified proteins of molecular weights similar to those expected for dimers and trimers of the target protein. Because the target protein appears to be located in the peripheral silica of the spicules, we named it "perisilin" (PSLIN).

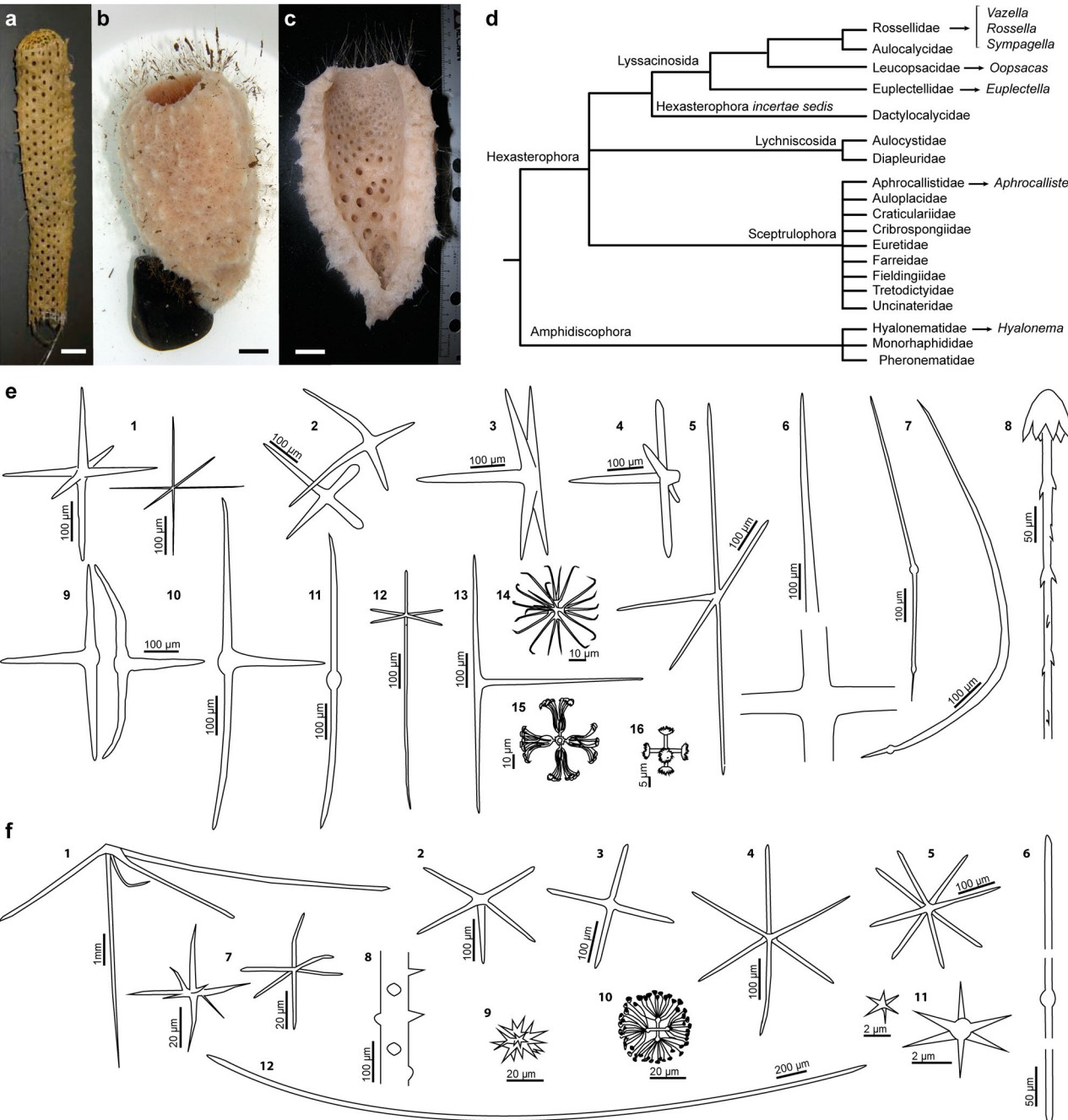

**Fig. 1 | Body shape, spicule complement, and relationships of the hexactinellid sponges *Euplectella curvistellata* and *Vazella pourtalesii*. a** *Euplectella curvistellata* voucher¸ republished from Shimizu et al.[33] with PNAS permission. **b, c** Live (**b**) and longitudinal-sectioned (**c**) voucher of *Vazella pourtalesii*. Scale bars (**b**–**d**): 2 cm. **d** Phylogenetic scheme of Hexactinellida illustrating the relationships of the genera hosting species with available genomic and transcriptomic data used in this study. Relationships are *sensu* Dohrmann[104], but updated according to the World Porifera Database https://www.marinespecies.org/porifera/, last visited on 13/01/23. **e** Schematic representation of spicules in *E. curvistellata*, as compiled from various sources[56,105–107], using Adobe Illustrator 2023 (version 27.1.1). 1: Parenchymalia-principalia hexactines; 2: Dermalia-choanosomalia stauractines; 3: Dermalia-choanosomalia pentactine; 4: Gastralia-canalaria pentactine; 5: Dermalia-choanosomalia paratetractine; 6: Large choanosomal stauractine; 7: Large choanosomal diactines of the sieve plate; 8: Anchorate basalia; 9: Dermalia-choanosomalia triactine; 10: Oscularia triactines; 11: Principalia diactine; 12: Dermal hexactine; 13: Choanosomal tauactine; 14: Oxyhexaster; 15: Floricome; 16: Graphicome. **f** Schematic representation of spicules in *V. pourtalesii*, as compiled from various sources[108,109] and personal skeletal observations, using Adobe Illustrator 2023 (version 27.1.1). 1: Hypodermal pentactine; 2: Dermal pentactine; 3: Dermal stauractine; 4: Atrial hexactine; 5: Atrial spicule with branching near central cross (tangential plane); 6: Choanosomal diactine; 7: Hexactines; 8: A fragment of tangential ray of hypodermal pentactine; 9: Hexaster; 10: Discohexaster; 11: Hexactines; 12: Prostalia monaxone.

Thus, hexaxilin emerges as the main component of the axial filament in both *E. curvistellata* and *V. pourtalesii* spicules, while glassin and perisilin, respectively, form concentric, network-like layers embedded in their peripheral silica of those species (Fig. 2r).

In addition to the identified proteins above described, SDS-PAGE and Western blot analyses detected several minor bands characterized by molecular weights higher than 70 kDa (Supplementary Fig. 1e). Protein abundance in these bands was systematically low and did not

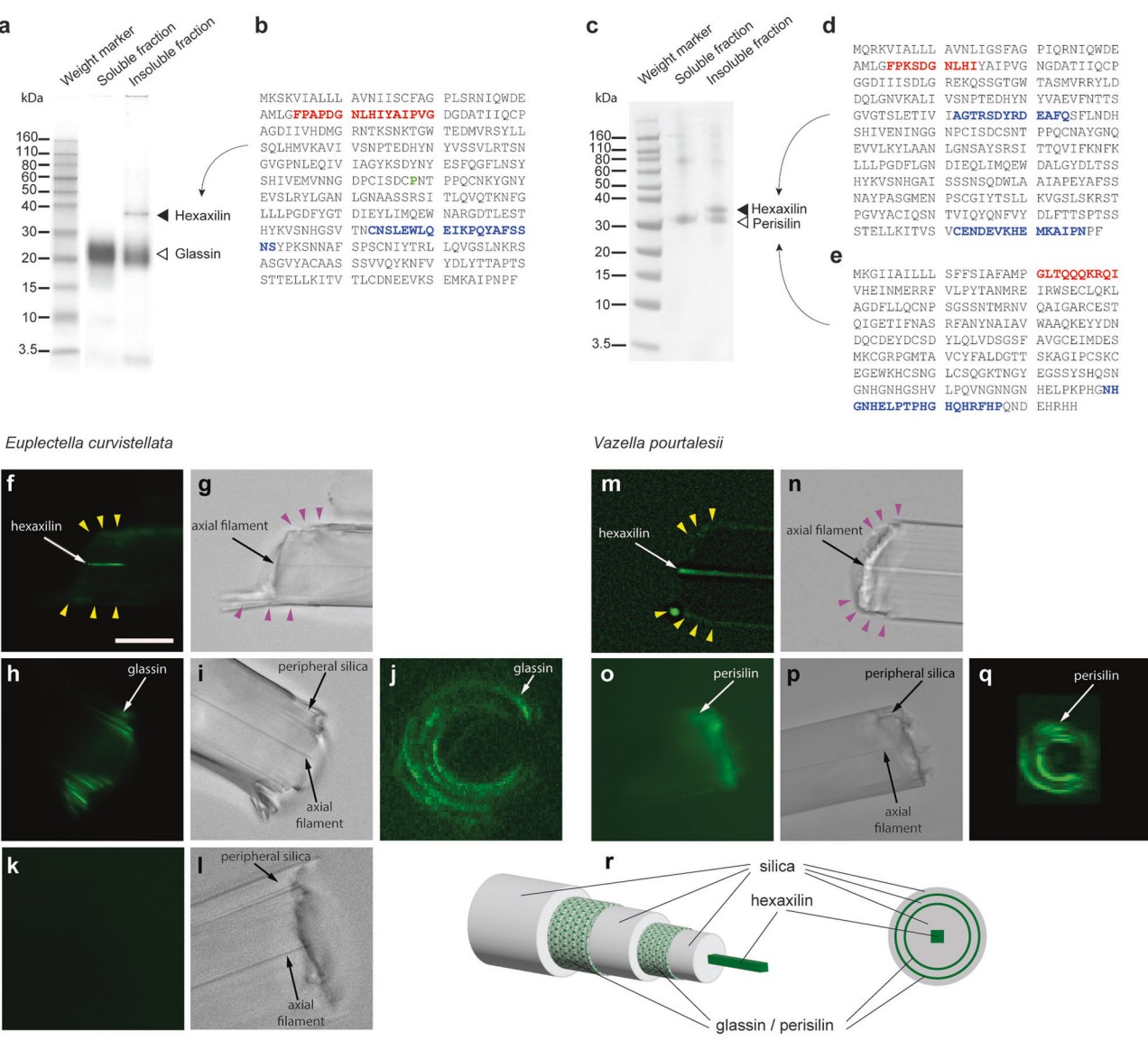

**Fig. 2 | Characterization and location of proteins occluded in the silica of the hexactinellids *Euplectella curvistellata* and *Vazella pourtalesii*. a** SDS-PAGE analysis of water-soluble and water-insoluble extracts from the silica of *E. curvistellata*, showing molecular-weight bands (kDa) for hexaxilin and glassin proteins. **b** Amino acid sequence of hexaxilin-1 obtained by translating the sequence of a PCR product amplified from *E. curvistellata* genomic DNA. **c** SDS-PAGE analysis of water-soluble and water-insoluble extracts from the silica of *V. pourtalesii*, *a*, showing bands of hexaxilin and perisilin proteins. **d**, **e** Amino acid sequence of hexaxilin-1 (**d**) and perisilin-1aα (**e**) from *V. pourtalesii* obtained by blast against *V. pourtalesii*'s transcriptome. Red letters in aa sequences represent peptides initially determined by the aa sequencer; blue letters indicate peptides used as immunogens for antibody production. **f**–**q** Fractured spicules incubated with rabbit antisera against hexaxilin, perisilin and glassin of *E. curvistellata* and *V. pourtalesii* for spatial localization of these proteins after labeling primary antibodies with Alexafluor 488-conjugated goat anti-rabbit IgG. Figures show incubations with: (**f**, **g**) anti-*E. curvistellata* hexaxilin antiserum; (**h**–**j**) anti-*E. curvistellata* glassin antiserum; (**k**, **l**) normal serum; (**m**, **n**) anti-*V. pourtalesii* hexaxilin affinity purified antibody; and (**o**–**q**) anti-*V. pourtalesii* perisilin affinity purified antibody. (**f**, **h**, **k**, **m**, **o**) fluorescent images; (**g**, **i**, **l**, **n** and **p**) phase contrast images. Images (**j**) and (**q**) are Z-stack micrographs of spicules in cross-section captured under the laser-scanning confocal microscope. Note in images (**f**) and (**m**) a very faint extra-axial staining (yellow arrows), which corresponds to spicule zones in images g and n (purple arrows) where the silica is irregularly fractured, creating microcavities from which the fluorochrome is rinsed with greater difficulty after the incubations, leaving traces of it that account for the very faint extra-axial color signal (see Supplementary Fig. 2). Scale bar in image (**f**) represents 50 μm and it applies to all images from (**f**) to (**q**). SDS-PAGE (**2a**, **2c**) and immunostainings (**2f**–**i**, **2k**–**p**) were repeated at least three times with similar results. **r** Schematic drawing made with Adobe Illustrator 2023 (version 27.1.1) representing a three-dimensional side view (left) and a cross section (right) of a spicule to summarize the spatial distribution of hexaxilin, glassin and perisilin.

allow any further protein identification through our Edman degradation approach. Future approaches using LC/MS-based proteomic methods might be more successful. Likewise, our gels (Supplementary Fig. 1a, c, e) never revealed bands compatible with actin (42 kDa), a protein recently reported by Ehrlich et al.[36,37] to operate as a template for the silicifying proteins of the axial filament of all three siliceous classes of Porifera. However, our methodological approach to identify silicifying proteins differs in several respects from that used in studies aimed at detecting actin (Supplementary Note 3). Therefore, our results cannot be interpreted as conclusive evidence to decide on the absence/presence of actin in the silica of *E. curvistellata*, *V. pourtalesii* or other hexactinellids.

| | Hyalonema populiferum | Aphrocallistes vastus | Euplectella curvistellata | Oopsacas minuta | Sympagella nux | Rossella fibulata | Vazella pourtalesii |
|---|---|---|---|---|---|---|---|
| BUSCO | 36.7% | 78.4% | 64.1% | 74.9% | 77.6% | 64.8% | 81.1% |
| HXX-1 | 1 | 1 | 1 | 1 | 1 | 1 | 1 |
| HXX-2 | | 1 | 1 | 1 | | 1 | 1 |
| HXX-3 | | 1 | 1 | 1 | 1 | 1 | |
| HXX-4 | | 1 | 1 | 1 | | | 1 |
| HXX-5 | | 1 | 1 | 1 | 1 | | 1 |
| HXX-5b | | 1 | | 1 | | | 1 |
| HXX-6 | | 1 | 1 | 1 | | | 1 |
| HXX-7 | | 1 | 1 | 1 | 1 | | 1 |
| HXX-8 | | 1 | 1 | 1 | | | 1 |
| PSLIN-1 | | | | 1 | 5 | 2 | 6 |
| PSLIN-2 | | | 1 | | | 1 | 10 |
| PSLIN-3 | | | | | | 7 | 2 |
| PSLIN-4 | | | | | | | 5 |
| PSLIN-5 | | | | | | | 2 |
| PSLIN-6 | 1 | 4 | 5 | 1 | | | 1 |
| PSLIN-7 | | | | 3 | 5 | | |
| PSLIN-8 | | 12 | | | | | |
| PSLIN-9 | 1 | | | | | | |
| PSLIN-10 | | 1 | | | | | |
| GLSIN | | 1 | 1 | 1 | 1 | 1 | 3 |

**Fig. 3 | Diversity summary for hexaxilins, perisilins and glassins in Hexactinellida.** BUSCO completeness (metazoans) of the hexactinellid genomes/ transcriptomes used in the present study is given in row 1, selecting for the most complete genome/ transcriptome if more than one were available to a species. Green and white cells indicate respectively the presence and absence of proteins. Numbers within green cells refer to number of different amino acid sequences found per protein in each species.

## Hexaxilin: evolution, expression patterns, and putative role in silicification

The genome of *E. curvistellata* revealed seven putative hexaxilin homologs (Supplementary Data 1), with identities from 43% to 38% and expected (E) values from $3.3e^{-79}$ to $4.8e^{-69}$. We named the characterized protein hexaxilin-1 and the related proteins as hexaxilin-2 to hexaxilin-8, following the order of E-values. The *V. pourtalesii* transcriptome revealed another eight hexaxilin sequences, seven of which (HXX-1, -2, -4, -5, -6, -7, -8) showed high pairwise identity (67.5%-81.5%) to their respective counterparts in *E. curvistellata* (Fig. 3, Supplementary Data 1). Although no HXX-3 occurred in *V. pourtalesii*, there was an additional sequence, HXX-5b, with an average identity of 47.5% to hexaxilin-5 of *V. pourtalesii* and *E. curvistellata*.

The search for hexaxilins was then extended to all hexactinellid species with genomes and transcriptomes in open sources, to a representative set of metazoans and two choanoflagellates, and, finally, to a general blast search at NCBI and EukProt (Supplementary Data 1). Additionally, hexaxilin homologs in distant taxa were searched analyzing the 3D secondary structure of proteins with Foldseek (Supplementary Data 2). Within Hexactinellida, data were available for only five additional species (*Rosella fibulata*, *Sympagella nux*, *Oopsacas minuta*, *Aphrocallites vastus*, *Hyalonema populiferum*), representing the two subclasses (Hexasterophora, Amphidiscophora), as detailed in Fig. 1d and Supplementary Note 4. In these hexactinellids, a variety of homologs of HXX-1 to HXX-8 occurred with bit scores ranging from 231 to 743, and averaging 309 (Fig. 3, Supplementary Data 1, Supplementary Fig. 3). Searches outside hexactinellid sponges retrieved only sequences with low identity (<35%), low bit scores (50 to 123; 87 on average), and comparatively high E-values (> $6.0e^{-31}$), casting reasonable doubt about the preservation of the original hexaxilin function (Supplementary Data 1). Thus, we nicknamed them hexaxilin-like (Supplementary Note 5).

Blast and Foldseek searches revealed hexaxilin to be phylogenetically related to metallo-beta lactamase (MBL) fold proteins (Supplementary Note 6), which include DNA internalization-related competence proteins (ComEC)[38] and enzymes for degradation of antibiotics in multidrug-resistant bacteria[39]. Some hexaxilin sequences are still reminiscent of the metal-binding sites of MBL fold (MBLf) proteins (Supplementary Fig. 3). Surprisingly, although MBLf proteins are widespread among multicellular eukaryotes[40,41], extensive searches yielded only five hits among unicellular eukaryotes, showing an unconventional character distribution across protists, which included only marine representatives. One is the Aplicomplexa gregarine *Siedleckia nematoides*, which parasitizes polychaetes. The four remaining hits belong to the supergroup Rhizaria (Cercozoa + Foraminifera + Radiolaria), that is, protists emitting pseudopodia. Three of them are Cercozoa (the chlorarachiophyte algae *Bigelowiella natans* and *Amorphoclora amoebiformis*, and the giant amoeba *Gromia sphaerica*). The fourth rhizarian is the foraminifer *Bolivina argentea*. Surprisingly, Radiolaria −the only silicifying rhizarians− and choanoflagellates lacked hexaxilin-like sequences.

Blast searches revealed hexactinellid hexaxilins to be also related to various previously unannotated−now hexaxilin-like−sequences of both other sponges (Demospongiae, Calcarea, Hexactinellida; Supplementary Fig. 4) and some bilaterians (Supplementary Data 1, Supplementary Note 5). To our surprise some of those bilaterians are not essentially silicifiers but can produce siliceous parts at a moment of their life cycle and also express membrane transporters for silicon. This is, for instance, the case of *Lingula anatina* (Brachiopoda, Lingulata), which expresses hexaxilin-like sequence Lan_013422193/ Lan_P037538 (bit score 76, identity of 24%) along with silicon transporter Lsi2/ArsB[15,21]. The co-occurrence of these two elements of the molecular machinery for dSi processing correlates with the fact that some Lingulata brachiopods (mostly family Discinoidea) produce an organosiliceous material that serves as shell for the larva[19]. Likewise, we have retrieved a hexaxilin-like sequence (CT_89866/CT_P002803; bit score of 97, identity of 24%) in the polychaete *Capitella teleta*, an organism that also expresses two types of silicon transporters (SIT-L and Lsi2/ArsB) and that could incorporate traces of silicon in some chetae[21].

The phylogenetic analyses of the information compiled above depict hexactinellid hexaxilins as a relatively diversified but monophyletic group, clearly separated from hexaxilin-like sequences of non-hexactinellid sponges and the rest of organisms (Fig. 4, Supplementary

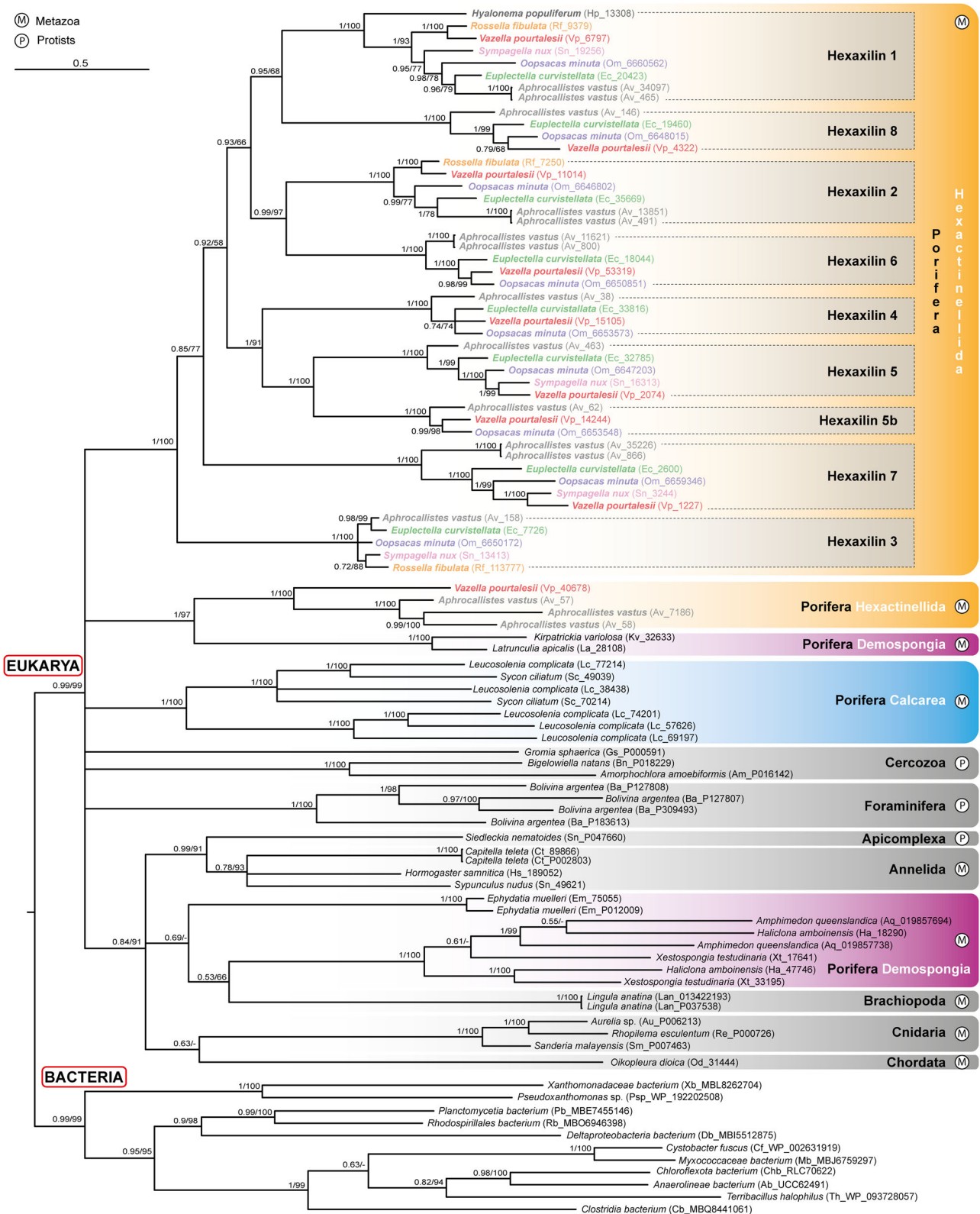

**Fig. 4 | Evolutionary relationships of *hexaxilin*.** Bayesian Inference (BI) phylogenetic tree of hexaxilins obtained with MrBayes 3.2. (Maximum Likelihood tree is shown in Supplementary Fig. 5). Included sequences correspond to all blast hits for hexaxilin-1 of *E. curvistellata* (sequence Ec_20423) with a bit score larger than 50 (Supplementary Data 1). Hexactinellid species are represented by names in different color. Scale bar represents 0.4 amino acid substitutions per site. Posterior probability (pp) and bootstrap (b) values are given at each node as pp/b. Nodes with no bootstrap value were not recovered in the Maximum Likelihood phylogeny. Alignment data are available in fasta format in Source Data Fig. 4.Source Data Fig. 4.

Fig. 5). Each hexaxilin type makes a well-supported, distinct clade, with hexaxilin-3 being the first one diverging. Hexaxilin-1 was the only clade including all six species of the subclass Hexasterophora along with the single representative of Amphidiscophora, which became the earliest diverging lineage. The smallest clade was hexaxilin-5b, showing that *V. pourtalesii*, *O. minuta*, and *A. vastus* have a hexaxilin type of their own (Figs. 3–4). The occurrence of HXX-2 to HXX-8 only in Hexasterophora and their absence in Amphidiscophora may reflect a phylogenetic pattern, but it may also be an artifact, since the quality of transcriptomes and genomes of the six Hexasterophora species is comparatively better (BUSCO completeness: 64–81%) than that of the transcriptome (36%) of the Amphidiscophora species (Fig. 3). The presence of several *hexaxilin* genes in most hexactinellid species may be related to the necessary production of axial filaments of different shapes to pattern the various spicule morphologies that characterize hexactinellid sponges (e.g., Fig. 1e, f). Comparatively, demosponges species have a smaller number of spicule types, and the different types are not produced by the participation of proteins from several unrelated genes, but only by silicatein isoforms, which have different expression patterns[42]. For instance, in the fresh-water demosponge *Ephydatia fluviatilis* the various isoforms are produced by expression of up to ten closely related *silicatein* genes[43].

It is worth noting that the hexaxilin-like sequences of the various protist phyla did not cluster together in the tree, being distributed in several independent subclades of hexaxilin-like sequences of demosponges and bilaterians, since the sister group of the hexactinellid hexaxilins emerged as a large, complex polytomy (Fig. 4, Supplementary Fig. 5). While the data available indicate that *hexaxilins* and *hexaxilin*-like genes are not a metazoan novelty, their scarce and "fragmentary" representation through protists is puzzling. Because hexaxilin-like sequences occur in Hexactinellida but also in Demospongiae and Calcarea (Supplementary Note 5, Supplementary Fig. 4), a likely hypothesis is that the hexaxilins of Hexactinellida with putative silicifying activity originated by duplication of non-silicifying hexaxilin-like genes of early Porifera inherited from protist ancestors and that further specialized in silica deposition only in Hexactinellida. Indeed, hexaxilin-like proteins have never been retrieved from extracts of the mineral skeletons of Demospongiae, Homoscleromorpha or Calcarea, either in this study or elsewhere[44,45].

A comparison of the hexactinellid hexaxilins with hexaxilin-like sequences of polychaetes and brachiopods indicates that they are all notably different, being bilaterian hexaxilin-like sequences more related to hexaxilin-like sequences of Demospongiae (Fig. 4, Supplementary Note 5, Supplementary Data 1, Source Data Fig. 4). Yet, while the brachiopod hexaxilin-like sequence shows long insertions at both the N- and C-terminal regions and lacks one of the six conserved cysteines characterizing hexactinellid hexaxilins, the polychaete hexaxilin-like protein has a sequence size in the range of hexactinellid hexaxilins and preserves all six cysteines. Because the exact mechanism and sequence domains through which hexaxilin polymerizes dSi remain to be deciphered, it is difficult to explain why the hexaxilin-like proteins of demosponges would not silicify, whereas the hexaxilin-like of brachiopods and other bilaterians would do it. It could be that independent mutational events in the "hexaxilin-like" MBLf inherited from protist ancestors might facilitate independent acquisition of silicifying capability in different branches of the metazoan lineage (Supplementary Note 5).

The expression of *hexaxilins* in *V. pourtalesii* individuals exposed to high dSi concentrations relative to that in control individuals revealed that *HXX*-1, -2, -5b, and -8 were significantly upregulated at high dSi (Fig. 5a, Supplementary Data 3–4). The fact that *HXX*-4, -5a, -6, and -7 were not does not necessarily mean that these genes are not also involved in silicification. Experiments with demosponges indicate that the axial filament of some spicule types is formed within the sclerocytes even under such low ambient dSi availability (0.5–1 μM) that it

barely allows incipient silica deposition around the filament[46,47]. In contrast, other spicule types are produced only when dSi is available at high concentration (e.g., 20 μM) and another types only if hyper-high dSi concentrations (100–250 μM) are provided[46–48]. Thus, non-upregulated *hexaxilins* would produce spicule types that are independent from ambient dSi availability.

Identical amino acids in the sequence of all hexactinellid hexaxilins include six cysteine residues (Supplementary Fig. 3) that may form disulfide bridges defining a common three-dimensional structure for these proteins and facilitating crosslinks with adjacent molecules. Calculations of the 3D secondary structure of hexaxilin-1 of *E. curvistellata* and *V. pourtalesii* with both SWISS-Model and Alphafold2 software come into general agreement, revealing abundance of alpha-helix, beta-turn, and sheet structures over the mature peptide (Fig. 6a, b, Supplementary Figs. 6–8). These features support that hexaxilins are globular proteins with a rigid 3D structure, having capability to build an axial filament. Thus, unlike traditionally assumed, the axial filaments of hexactinellids (made of hexaxilin) and demosponges (made of silicatein) would not be homologous, but analogous structures evolved to perform similar functions (Supplementary Note 7). Because the homosclerophorid sponges lack silicatein and hexaxilin, the siliceous skeletal parts of Hexactinellida, Demospongiae, and Homoscleromorpha emerge as non-homologous structures. This scenario is in line with an earlier hypothesis proposing that the ability to silicify could have been independently acquired more than once during the evolution of Porifera[49,50]. Such initial proposal was based on the realization of distinctive ultrastructural features of spicules and silicifying cells of Homoscleromorpha[49,51]. Subsequent studies confirming the absence of silicatein in Hexactinellida and of glassin in Demospongiae also support such a hypothesis[28,30,31]. Our current findings validate and expand that minority early view.

## Perisilin: evolution, expression patterns, and putative role in silicification

The transcriptome of *V. pourtalesii* revealed 26 transcripts homologous to *perisilin*, belonging to 10 genes with identities ranging from 98% to 29% and E-values from 0 to 7.64e[-10] (Supplementary Data 5). The characterized protein, which has three isoforms (α, β, γ), was named perisilin-1. However, to name the homologs as perisilin-2, -3, etc., following the order of E-values, was impractical because of high numbers of transcripts with similar E-values.

Blast searches for perisilins in hexactinellids other than *V. pourtalesii* retrieved 54 unannotated sequences with moderate to low identity (77–28%) and bit scores ranging from 50 to 255 (Supplementary Data 5). Out of Hexactinellida, another 105 matches arose, most of them in Metazoa, spanning from demosponges and calcareous sponges (Supplementary Fig. 9) to vertebrates (Supplementary Data 5). There were also six hits in protists. Foldseek searches for distant homologs (Supplementary Data 6) retrieved no relevant information additional to that of blast searches at NCBI and EukProt databases. The protist set consisted of the Amebozoa *Sappinia pedata*, the Ciliophora *Furgasonia blochmanni*, the Foraminifera *Nonionella stella* and *Bolivina argentea*, the Cryptista *Roombia truncata* and the Hemimastigophora *Hemimastix kukwesjijk*. Because they all were poorly related to perisilin-1 of *V. pourtalesii* (32–24% identity; bit scores <80, averaging 56), we designated them as "perisilin-like". As in the case of hexaxilin, radiolarians and choanoflagellates, despite being silicifying protists, did not contain any perisilin-related sequences. No recognizable homologs were found in Prokaryota either.

Phylogenetic inference revealed that hexactinellid perisilins make a well-supported monophyletic group, the major clades of which were named as perisilin-1 to perisilin-10 (Fig. 7, Supplementary Fig. 10). The sister group of perisilins is proposed to consist of only some perisilin-like proteins from freshwater demosponges (*Ephydatia muelleri*).

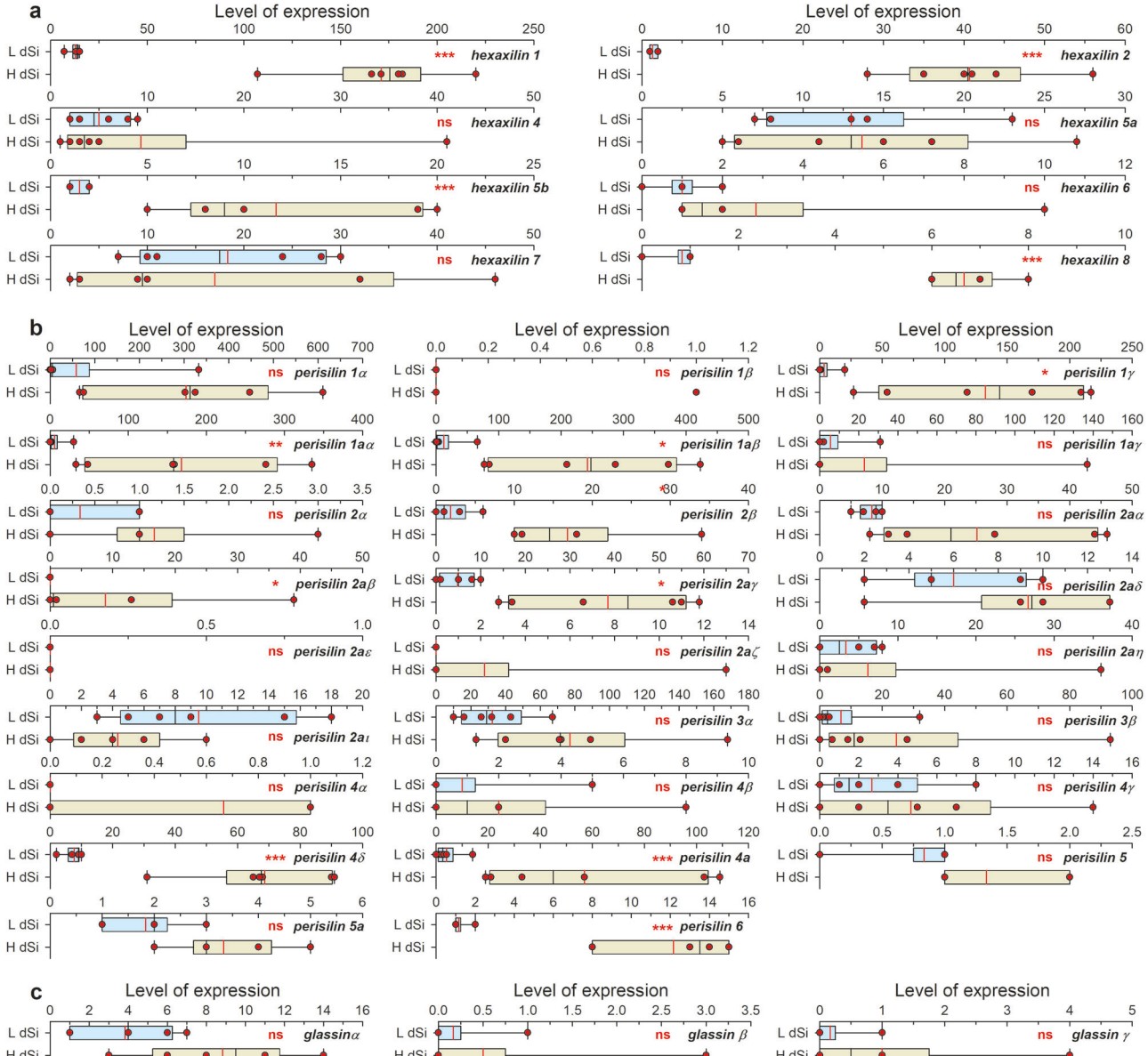

**Fig. 5 | Expression patterns of hexactinellid silica proteins.** Box plot of TMM normalized expression values (red dots) of *hexaxilin* (**a**), *perisilin* (**b**), and *glassin* (**c**) genes in the transcriptome of six individuals of *V. pourtalesii* exposed to naturally low dSi concentrations (light blue) versus that of six others exposed to high dSi concentrations (light brown). End of boxes define the 25th and 75th percentiles, with a black line at the median, a red line at the mean, and errors bars defining the 10th and 90th percentiles. Differentially expressed (DE) genes were determined through the quasi-likelihood *F* test implemented in the function GLMTreat of the Bioconductor package edgeR. Asterisks indicate the statistical significance of tests, following the criterion of at least a fourfold difference in expression and with the *P*-values corrected by false discovery rate (FDR): \*\*\**P* < 0.001; \*\**P* < 0.01; \**P* < 0.05; ns= not significant. TMM expression values per individual, statistical tests, and exact probabilities of tests are detailed in Supplementary Data 3–4.

Nevertheless, the node receives unreliable support, as does most of the phylogenetic structure of the outgroup, which is essentially a polytomy that leaves the phylogenetic origin of hexactinellid perisilins poorly resolved.

Perisilins are distantly related (identity <32%) to proteins in the CAP superfamily[52] of multicellular eukaryotes (Supplementary Data 5, Supplementary Note 8), which includes cysteine-rich secretory proteins, antigen 5 proteins, and glioma-pathogenesis-related (GLIPR) proteins[52]. Most CAP proteins are secreted to operate extracellularly[52]. Secretion and extracellular activity during the peripheral silica deposition is also the most plausible action pathway for putatively silicifying perisilins. A sequence analysis of the extracted perisilins (and also glassins) using Phobious revealed that both proteins have signal peptides (1-18 aa) of conventional structure (Supplementary Table S1), which would allow them to be exported through the plasmalemma of the scleroblasts for extracellular silicification. Such an action mode would explain not only how perisilins (and glassins) end embedded in the peripheral silica of hexactinellid spicules, but also how spicules that are orders of magnitude larger than the silicifying cells can be completed (see section The silicification process).

It is worth noting that none of the seven hexactinellid taxa studied has all ten perisilins (Figs. 3, 7). Perisilin-6 occurs in five species from five different families, but most other perisilin types occur in only one, two or three related species. This distribution pattern suggests that perisilins would be involved in producing skeletal features restricted to groups of related species (e.g., genus, family), making those groups skeletally distinguishable from others. However, while our data supports the putative role of perisilin in

**a** *E. curvistellata* Hexaxilin-1

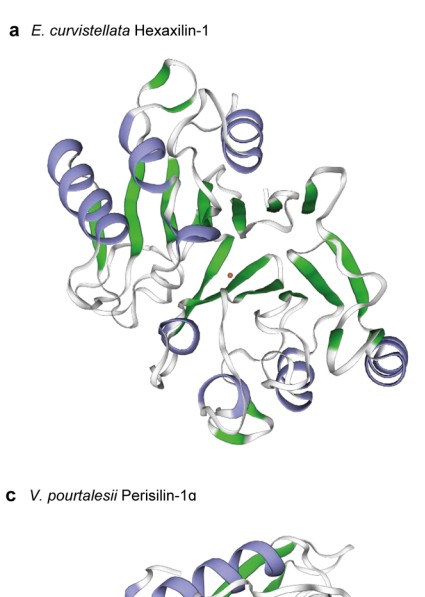

**b** *V. pourtalesii* Hexaxilin-1

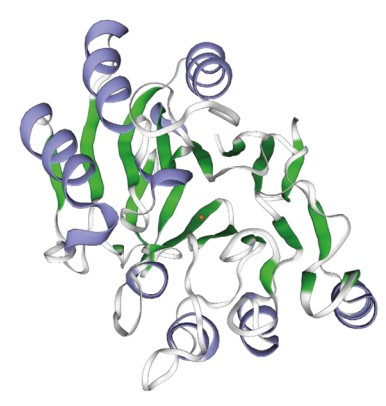

**c** *V. pourtalesii* Perisilin-1α           **d** *E. curvistellata* Glassin

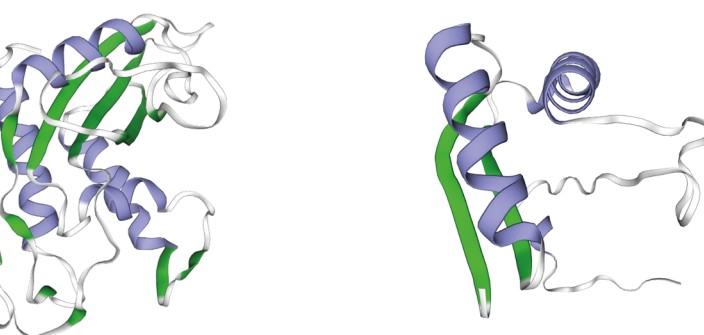

**Fig. 6 | 3D models for the secondary structure of proteins extracted from the silica of the hexactinellid species *Euplectella curvistellata* and *Vazella pourtalesii*.** Models were inferred with both SWISS-MODEL and Alphafold2 software, which came into general agreement (Supplementary Fig. 7, 8, 12, 15). Protein alignment and templates for SWISS-MODEL are shown in Supplementary Fig. 6. SWISS-MODEL results are shown because they are more conservative, not considering those regions that are not available in the templates. Alphapfold2 generated full-length polypeptides, hypothesizing about the regions not covered by the templates and suggesting that those regions not inferred by SWISS-MODEL were basically random structures. **a** Hexaxilin-1 of *E. curvistellata* (ranging from aa 41 to aa 308). **b** Hexaxilin-1 of *V. pourtalesii* (ranging from aa 41 to aa 315). **c** Perisilin-1α of *V. pourtalesii* (ranging from aa 21 to aa197). **d** Glassin of *E. curvistellata* (ranging from aa 21 to aa 197). Sequences expected to form alpha-helices and beta-sheets are colored in blue and green, respectively.

silicification, it is not clear whether all identified hexactinellid perisilins can silicify. Hexactinellid perisilins share a relatively conserved Cys-rich, GLIPR1-like domain reminiscent of CAP proteins (Supplementary Note 8). Such a domain is likely unrelated to their putative silicifying ability, which would seemingly emanate from His-rich regions at the C-terminus of perisilins (Supplementary Fig. 11), given that similar His-rich and Asp-rich regions occurring in glassin (but at the N-terminus) are known to promote silica deposition[33,34]. Nevertheless, the C-terminus of perisilins is so variable that His-rich regions can even be missing. Perisilins bearing His-rich domains that would putatively confer silicification ability occur in *V. pourtalesii* (PSLIN-1 to -4), *O. minuta* (PSLIN-7), and *S. nux* (PSLIN-1). In contrast, *V. pourtalesii* PSLIN-5 and 6, *O. minuta* PSLIN-6, *S. nux* PSLIN-7, and all perisilins of *H. populiferum*, *A. vastus*, *E. curvistellata*, and *R. fibulata* lack the His-rich regions. Identical amino acids of all hexactinellid perisilins include 12 cysteine residues (Supplementary Fig. 11), which, through disulfide bridges, may both define a conserved three-dimensional structure and facilitate crosslinks with adjacent molecules for assembling a protein sheet on the spicule surface for extracellular silicification. The secondary structure inferred for perisilin-1 of *V. pourtalesii* suggests that their GLIPR1-like domain appears to have some degree of 3D structure (Fig. 6c, Supplementary Fig. 12), while the His-rich domain at the C-terminus is flexible, lacking 3D structure. Flexible structures and random distribution of His in these proteins are favorable for interaction with a variety of chemical dSi species.

When the expression of *perisilin* genes of *V. pourtalesii* was examined, it tended to be higher in individuals exposed to high dSi concentrations, even if not always with statistical significance (Fig. 5b). Eight *perisilin* genes were statistically upregulated, including *pslin-1γ*, which encodes the protein extracted from the silica (Fig. 5b, Supplementary Data 3–4). Unlike for hexaxilins, the range of expression varied drastically between *perisilin* types, isoforms included. For instance, *perisilin-1α* was expressed in the hundreds range, while *perisilin-1β* was in levels below one. Thus, factors other than environmental dSi concentrations are expected to regulate *perisilin* expression.

Taken together, the tendency of perisilins to increase in expression with high dSi availability, their presence in the peripheral silica of spicules, and the His-rich domains of their sequence collectively support that perisilin is involved in silica deposition at least in *V. pourtalesii*, and probably also in *O. minuta* and *S. nux*.

## Glassin: evolution, expression patterns, and role in silicification

Glassin was originally described as a tandem repeat carrying three domains: (1) histidine-rich and aspartic-acid-rich (HD) domain, (2) proline-rich (P) domain, and (3) histidine-rich and threonine-rich (HT) domain[34]. The HD domain was demonstrated to be crucial for silicification, particularly in cooperation with the HT domain[34]. Since its discovery in *E. curvistellata*[33], the sequence of glassin (LCO10923.1; 233 aa) remained incomplete. Herein, a 2/3rd longer glassin sequence (Contig 45319; 398 aa) has been identified from that same genome, but

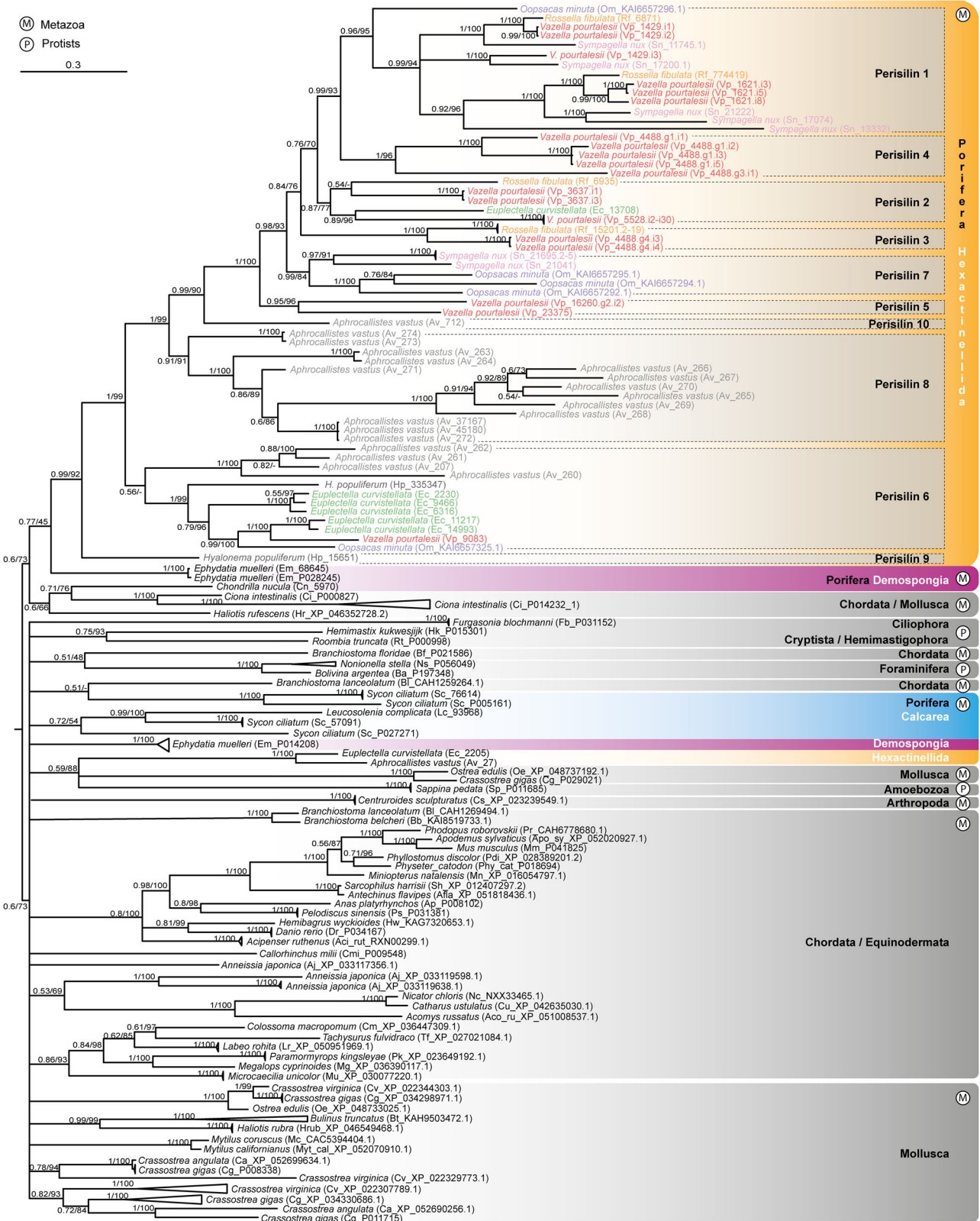

**Fig. 7 | Evolutionary relationships of *perisilin*.** Bayesian Inference (BI) phylogenetic tree of perisilins obtained with MrBayes 3.2. (Maximum Likelihood tree is shown in Supplementary Fig. 10). Included sequences correspond to all blast hits for perisilin-1α of *V. pourtalesii* (sequence Vp_1621.i8) with a bit score larger than 50, but very similar sequences for a given species in the ougroup have been collapsed in tree for the sake of clarity. All sequences are given in Supplementary Fig. 10 and Supplementary Data 5. Different hexactinellid species are represented by names in different color. Scale bar represents 0.3 amino acid substitutions per site. Posterior probability (pp) and bootstrap (b) values are given at each node as pp/b. Alignment data is available in fasta format in Source Data Fig. 7.Source Data Fig. 7.

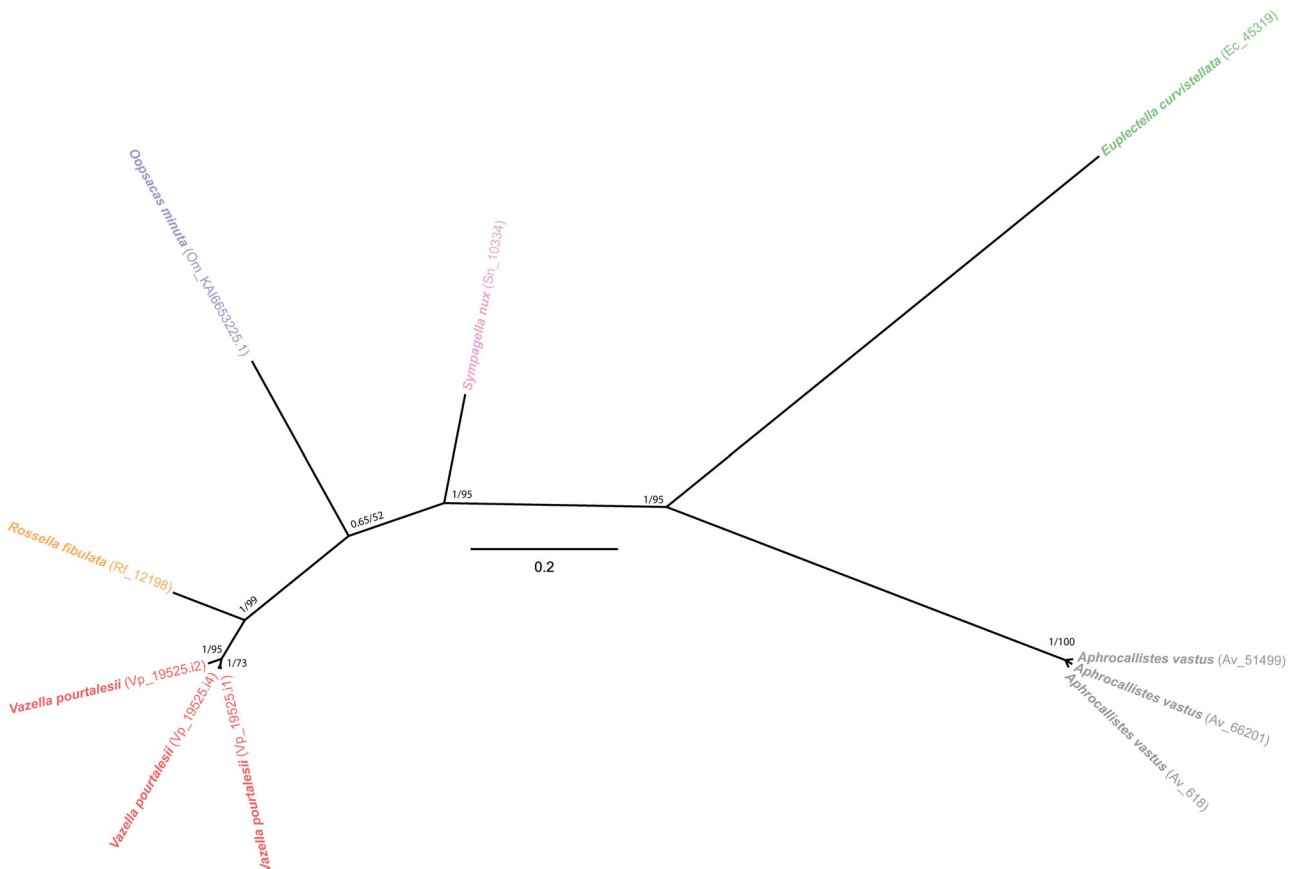

**Fig. 8 | Evolutionary relationships of *glassin*.** Unrooted Bayesian Inference (BI) phylogenetic tree of glassin obtained with MrBayes 3.2. (Maximum Likelihood tree is shown in Supplementary Fig. 13). Included sequences correspond to all blast hits for glassin of *E. curvistellata* (sequence Ec_45319) with a bit score larger than 50 (Supplementary Data 7). Each hexactinellid species is represented by a different color. Scale bar represents 0.2 amino acid substitutions per site. Posterior probability (pp) and bootstrap (b) values are given at each node as pp/b. Alignment data is available in fasta format in Source Data Fig. 8.Source Data Fig. 8.

it is also incomplete. It contains a single unit of the three domains and an uncharacteristic C-terminal sequence. The new sequence, when used as query, retrieved a homolog with three isoforms (40% to 55% identity) from the *V. pourtalesii* transcriptome, and only five additional homologs from the rest of organisms (bit scores from 114 to 393), all belonging to hexactinellids in the subclass Hexasteropohora (Supplementary Data 7). Only one sequence occurred in each of the rossellid species *R. fibulata*, *S. nux*, and *O. minuta*, while *A. vastus* had three, which are likely identical at the amino acid level (Fig. 3). Though taxon sampling is still limited, such a restricted distribution across taxa suggests that, unlike perisilin, glassin has only diversified among, but not even within, Hexasterophora species.

As no reliable ancestral or derived gene (i.e., bit score » 50) is retrieved for glassin out of Hexasterophora hexactinellids by blast searches and Foldseek searches, the origin of this protein remains enigmatic and its phylogeny unrooted. Phylogenetic inference (Fig. 8, Supplementary Fig. 13) shows glassins of the rossellids *V. pourtalesii* and *R. fibulata* being more closely related to that of the leucopsacid *O. minuta* (low-support node) than to that of a third rossellid, *S. nux*. Glassins of *E. curvistellata* and *A. vastus* are highly differentiated not only from those in the rest of taxa, but also from each other. The best conserved region of glassin in *E. curvistellata* and *V. pourtalesii* involves mostly the non-silicifying C-terminus (position 277 to 476 in the alignment; Supplementary Fig. 14). Calculations of the secondary structure of glassin of *E. curvistellata* predicted some 3D structure for the C-terminal sequence (Fig. 6d, Supplementary Fig. 15), but not for the HD, P, and HT domains related to the silicification function. Glassin of *V. pourtalesii* shows an N-terminus that, despite not aligning well with that of glassin of *E. curvistellata*, it is still very rich in His and Asp, suggesting that glassin of *V. pourtalesii* might silicify as well. In contrast, glassins of *A. vastus*, *R. fibulata*, and *S. nux* lack those His-rich regions, suggesting no silicifying capability. Nevertheless, the global picture is still blurry, since glassin could not be extracted from the same silica of *V. pourtalesii* that provided perisilin, and a specific antibody developed herein for *V. pourtalesii* glassin (Supplementary Fig. 14; see Methods) produced no silica tagging. Additionally, none of the three *glassin* isoforms in *V. pourtalesii* was upregulated in high dSi (Fig. 5c, Supplementary Data 3–4). Thus, part of the evidence suggests that glassin would silicify in euplectellids but not in rosellids, which would rather use perisilins.

With the available information, the most plausible hypothesis is glassin being an acquisition of the hexactinellid ancestor of Hexasterophora. It would have subsequently diversified in this subclass, becoming specialized for silicification in only some hexasterophoran lineages, where it participates in silica deposition processes that facilitate skeletal differentiation between species but not within them. That latter role would rather have been played by hexaxilins and perisilins.

### The silicification process in Hexactinellida
TEM observations were conducted on both silicified (Fig. 9a, b) and desilicified (Fig. 9c–f) tissue sampled of three *V. pourtalesii* individuals that were incorporating dSi at high rate during the high dSi treatment[15] (Methods). In silicified samples, the silica and the axial filament of the spicules were dragged away when the diamond blade cuts through the spicules. Therefore, only tiny spicules (or early

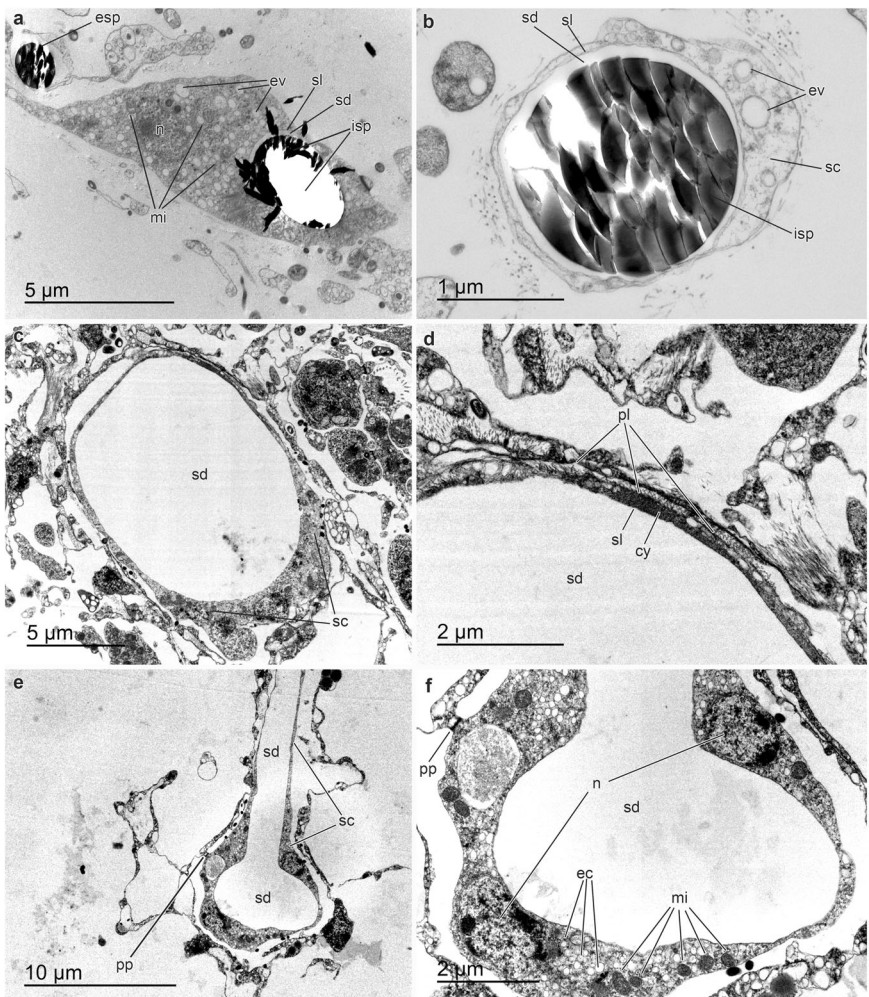

**Fig. 9 | TEM micrographs of silicification in *Vazella pourtalesii*. a, b** Cross-sections of early-developing stages of spicules (isp) still within the silica deposition vesicle (sd) in the cytoplasm (cy) of the sclerocytes (sc). The vesicle is limited by the silicalemma (lm), a membrane distinct from the plasmalemma (pm) of the sclerocyte. In these silicified ultra-thin sections, the silica (seen as an electron-dense material) of the intracellular (isp) and extracellular (esp) spicules has been crushed into small pieces by the diamond blade and dragged away. The hexactinellid sclerocytes show nucleated nuclei and are rich in electron-clear vesicles (containing dSi?) and mitochondria, being these features similar to those in sclerocytes of Demospongiae. **c–f** Desilicified samples showing large empty holes within the silica deposition vesicle (sd) of the sclerocytes (sc) in which spicules where hosted prior to desilicfication in HF. Image (**d**) is a magnification of image c detailing an electron-dense zone of the sclerocyte cytoplasm (cy) adjacent to the silicalemma (sl), as also reported in previous literature[54]. Image f details a longitudinal section of the space left by a large desilicified spicule that was enclosed within a sclerocyte (sc). Two nuclei (n) are seen in this sclerocyte section (i.e., sclerosyncytium). The sclerosyncytium is rich in mitochondria (mi), electron-clear vesicles (ev), and is connected to other cells through perforated plugs (pp). Findings derive from inspection of tissue from three different sponge individuals that were randomly collected but showed high silicate consumption rates during the high dSi treatment.

developing stages) were sectioned and studied properly (Fig. 9a, b). Such spicules occurred in sclerocytes, within a silica deposition vesicle limited by a silicalemma membrane (Fig. 9a, b). Like in demosponges and homosclerophorids, silicifying "cells" of *V. pourtalesii* contain many electron-clear vesicles (transporting dSi?) and high number of mitochondria, attesting that silicification requires substantial energy investment. Unlike in demosponges, hexactinellid silicifying "cells" are multinucleated (i.e., sclerosyncytia) and their cytoplasm is connected to that of other sclerosyncytia—also that of another cell types—through perforated intercellular plugs[53] (Fig. 9e, f) to form large silicifying units.

From our results and the scarce available literature[54–58], the predicted sequence of steps in the making of hexactinellid spicules involves the initial intracellular production of an axial filament of hexaxilin within the silica deposition vesicle of sclerosyncytia. Hexaxilin-1 of *E. curvistellata* and *V. pourtalesii* is predicted by Phobius[59] web server (https://phobius.sbc.su.se/) to have a signal

peptide (1-19 aa), which is consistent with its proposed translocation from the cytosol to the silica deposition vesicle for the formation of the axial filament necessary to initiate intracellular silica deposition (Supplementary Table S1). We assume that different hexaxilins would assemble axial filaments with different number of axes for patterning different spicule shapes. Once hexaxilin becomes surrounded by silica, the primordial spicule would be exocytosed for glassins and/or perisilins to guide extracellularly the deposition of the peripheral silica during thickening and/or ornamentation. The silicifying activity of perisilin and glassin appears restricted to only some lineages of Hexactinellida, remaining unclear whether these two proteins can cooperate in some of those lineages to produce the peripheral silica. In Demospongiae, the final silicification steps of large spicules are also suspected to occur extracellularly[60] and several mechanisms have been hypothesized[61–63] that differ from the multi-protein scenario herein uncovered for hexactinellids (Supplementary Note 9). Yet, it might be that not all hexactinellid spicules are finished extracellularly,

because complete intracellular formation has also been postulated for relatively large spicules[54].

Although silicification in Hexactinellida appears to involve at least three unrelated silicifying proteins, glassin and perisilin appear to operate restrictively and in different lineages. Thus, we predict that additional silicifying proteins, functionally analogous to glassin and perisilin but pending characterization, may participate in the silicification of understudied hexactinellid lineages. It is also worth noting that hexaxilin, perisilin, and glassin have experienced contrasting patterns of diversification within Hexactinellida, facilitating the amazing skeletal diversity that characterize this sponge class, compared to that of Demospongiae and Homoscleromorpha. As we propose that hexaxilin, perisilin and glassin guide silicification in Hexactinellida, while silicatein does it in Demospongiae, and none of these four proteins operates in Homoscleromorpha, it appears that all three classes of siliceous sponges utilize independent protein machineries for building their siliceous skeletons, which emerge as nonhomologous structures.

The emerging scenario, in which we predict multiple unrelated silicifying proteins (hexaxilin, perisilin, glassin, and likely others still uncharacterized) to be responsible for generating the skeletal diversity of Hexactinellida, invokes an "evolutionary strategy" radically opposite to that in Demosponges, also to that in other efficient silicifiers, such as diatoms. Skeletal diversity in Demospongiae is not generated through involvement of several unrelated silicifying proteins (Supplementary Note 9), but through just silicatein evolving various isoforms[28], with different combinations of isoforms and their distinct expression patterns producing different spicule types[42,43]. Likewise, the silicification system of diatoms is based on small silaffin peptides, characterized by many repeat units rich in Lys-Lys pairs that link both to methyl groups and to long-chain polyamines (LCPAs), and that may further get phosphorylated[64]. Such post-transcriptional modifications of the original peptide skeleton trigger a combination of cationic and hydrogen-bonding interactions that facilitate binding to the surface of silica particles and dSi polymerization[65–67]. Thus, different combinations of silaffin and LCPAs generate potential for nearly "infinite" diversity in diatom skeletal morphology. It remains unknown whether analogous post-transcriptional modifications to favor dSi polymerization and skeletal diversity might occur in the set of silicifying proteins known in Porifera.

## Origin and evolution of silicification
The fact that the three classes of siliceous sponges appear to have evolved independent protein systems for dSi polymerization helps to clarify the Precambrian scenario in which Porifera appeared and diversified. There is plenty of evidence that Demospongiae and Hexactinellida lineages had already diverged and diversified by the Late Cambrian (497 mya)[68–70]. However, there is currently an important time lag between the Cryogenian origin estimated for Metazoa and Porifera by molecular clocks (850-650 mya)[69,71,72], the oldest steroid markers for sponges[73,74] (660-635 mya), and the occurrence of unambiguous sponge fossils (<535 mya). The oldest unambiguous hexactinellid fossils date from the earliest Cambrian (535 Ma) at Elburz Mountains, Iran[75]. The fossil origin of demosponges is best marked by the appearance of archaeocyathan remnants (c. 526 Ma)[75]. The information about the origins of Homoscleromorpha is even more scarce and difficult to interpret (Supplementary Note 10).

Studies on marine cherts[76,77] suggest that the concentration of dSi in Precambrian and Cambrian oceans was near saturation (i.e., around 1500 μM)[78,79]. Consequently, early oceans provided a much higher dSi availability to potential silicifiers than the modern ocean, which is characterized by an average dSi concentration of about only 1 μM in photic zone and less than 10 μM on a global scale[80]. Thus, sponges appeared and started diversifying in an ocean highly rich in dSi (also colder, with lower oxygen levels, and different redox conditions than the modern ocean[81]). At the very high dSi concentrations of those ancient oceans, dSi is predicted to enter the sponge cells by passive transport in favor of concentration gradient through aquaglyceroporin channels in the cell membranes[15]. Thus, early sponges likely needed to actively expel dSi from their cells to keep homeostasis, a function that is thought to be carried out by ArsB silicon transporters[15]. Both passive aquaglyceroporins (dSi influx) and active arsB transporters (dSi efflux) are ancestral homeostatic mechanisms widely distributed across modern prokaryotic and eukaryotic lineages for transmembrane transports of small solutes, including silicon in the case of Porifera[14,15,21].

It is worth noting that whereas Demospongiae and Hexactinellida share related aquaglyceroporins and arsB transporters (i.e., inherited from a putative ancestral sponge) to deal with dSi transport across cell membranes[15], these sponge classes have evolved independent protein mechanisms for dSi polymerization. These two contrasting features can only be reconciled by considering that 1) sponges would have appeared and diverged into Demospongiae and Hexactinellida during the Precambrian (Cryogenian-Edicaran, 850–635 mya), as indicated by molecular clocks, and 2) that those Precambrian demosponges and hexactinellids would not yet be able to polymerize dSi into silica, and would not leave behind recognizable traces of mineral skeleton, as indicated by the fossil record. Therefore, Cryogenian-Edicaran demosponges and hexactinellids coped with high dSi concentrations only by expelling dSi from their cells and syncytia through arsB transporters. However, through evolution, those early non-skeletonized sponge lineages evolved more efficient strategies to remove harmful dSi from the cytoplasm of cells. In addition to just continuously expelling it through arsB transporters (which consume ATP), sponges started to inactivate the intracellular dSi by precipitating it, giving rise to the ability of biosilicification. Such a mechanism to get rid of the high intracellular dSi concentrations, over the course of evolution, would have become part of the sponge skeletogenesis, likely favored by energetics and by providing new potential for increasing body size. As Demospongiae and Hexactinellida lineages would have diverged molecularly and cellularly before the rising of silica production, their mechanisms to polymerize dSi (i.e., silicifying protein machineries) arose independently and at different geological times, as indicated by the fossil record (Supplementary Note 10). A very similar evolutionary scenario was proposed by Ma and Yang[72] on the independent basis of a molecular clock study, which suggested Demospongiae and Hexactinellida to start silicifying only after their divergence. The idea of the Poriferan classes diverging time before the emergence of silicification reconciles not only the current mismatch between the messages of the molecular clocks and the fossil record, but it also explains why silicifying proteins would have evolved independently in the siliceous classes whereas silicon transporters did not.

## Methods
### Sponge samples
Specimens of *Euplectella curvistellata* were collected by beam trawling at a depth of 236 m at 32°30′ N, 129°10′ E in the East China Sea on March 4, 2012, during an expedition (Expedition 343) on the research vessel Nagasaki Maru of Nagasaki University (Japan).

Specimens of *Vazella pourtalesii* were obtained during an oceanographic mission on the Canadian Coast Guard Ship Martha L. Black from 2 to 7 September 2017, funded by the European Union–H2020 SponGES grant and aimed to investigate the North-Atlantic aggregations that *V. pourtalesii* forms in Sambro Bank Sponge Conservation Area on the Nova Scotia continental shelf (Canada). Collections of live individuals were conducted using the remotely operated vehicle ROPOS", owned and operated by the nonprofit Canadian Scientific Submersible Facility (CSSF, Canada). Collections took place at Emerald Basin (43°52′ N, 6302′ W; 207-215 m deep).

Collection and experimentation with the mentioned sponge species complied with all ethical regulation and permits under national and international legislations.

## Extraction and identification of proteins from sponge silica

Extracts from the silica skeleton of *E. curvistellata* and *V. pourtalesii* were prepared as previously described by Shimizu et al.[33]. Briefly, dry sponge tissue was digested in a commercial bleach solution, then in $HNO_3/H_2SO_4$ (1:4) to further eliminate any organic trait outside the silica skeleton. The organic material-free silica skeleton was immersed $HF/NH_4F$ (1:4) for about 2–3 days, until silica was dissolved. The solution was dialyzed against $H_2O$ using dialysis tube (SnakeSkin Dialysis Tubing, 3.5 K MWCO, Thermo Fisher Scientific). The dialysate was centrifuged to separate the supernatant and the precipitate, which was subjected to SDS/PAGE analysis using NuPAGE 4-12 % Bis-Tris Pre-Cast gels (Thermo Fisher Scientific) with MOPS SDS running buffer (Thermo Fisher Scientific). Samples were then dissolved in 1x LDS sample buffer (Thermo Fisher Scientific) containing 1% DTT. After electrophoresis, the gels were stained with CBB (EzStain Aqua; ATTO) for protein visualization. Molecular masses of the proteins were estimated using Novex Sharp Prestained Protein Standards (Thermo Fisher Scientific).

For Western blot analysis, proteins separated by SDS/PAGE were electrophoretically transferred to polyvinylidene difluoride (PVDF) membranes (ATTO). Molecular masses of proteins in electrophoresis and Western blotting were estimated using MagicMark XP Western Protein Standard (Thermo Fisher Scientific) as markers. Proteins were detected with WesternBreeze Chromogenic Western Blot Immuno-detection Kit, anti-rabbit (Thermo Fisher Scientific), which was conjugated (according to manufacturer's instructions) to primary rabbit polyclonal antibodies developed against the proteins of interest through Sigma-Aldrich Japan (Tokyo, Japan). For antibody production, short peptides (ranging from 13 to 20 aa) of the proteins of interest were selected as epitopes. For *E. curvistellata*, we selected the sequence CNSLEWLQEIKPQYAFSSNS, located at positions 253-272 of hexaxilin-1(Fig. 2b) and the sequence HGKHGKHGKHDHHDHHH, located at positions 99–116 of glassin (Supplementary Fig. 14). For *V. pourtalesii*, we selected sequences AGTRSDYRDEAFQ (not successful) and CENDEVKHEMKAIPN (successfully working), located at positions 132-140 and 342-356 of hexaxilin-1, respectively (Fig. 2d), sequence NHGNHELPTPHGHQHRFHP, located at positions 239-257 of perisilin-1α (Fig. 2e), and sequence PEDAGRSELIEDVT, located at positions 364-377 of glassin transcript Vp_19525.i4 (Supplementary Fig. 14).

To mark glassin and hexaxilin of *E. curvistellata* on the gels, the primary antibody against Ec glassin (1: 1,000 dilution) and Ec hexaxilin (1: 1,000 dilution) were used in the form of antisera. In contrast, antisera containing anti-hexaxilin and anti-perisilin of *V. pourtalesii* were affinity-purified with protein-G spin columns, and then used as primary antibodies (1: 100 dilution) for detection of Vp hexaxilin and Vp perisilin on gels, respectively. As also reported in previous literature[82], our antibodies performed similarly well, irrespective of being used in serum or affinity purified. The immunolabelling of the proteins of interest in gels provided tests of the adequate specificity of the antibodies.

For analysis of the amino-terminal peptide sequences, the water-insoluble fraction was separated by SDS/PAGE, then electrophoretically transferred to polyvinylidene difluoride (PVDF) (ATTO) membranes. After CBB staining, the PVDF membranes containing the putative proteins of interest were processed on an automated Edman-degradation protein sequencer (PPSQ31A; Shimadzu).

The gene encoding a 38 kDa protein was identified by the tblastn option of ncbi-blast-2.12.0⁺ software, using the amino-terminal sequence of 38 kDa protein as a query and whole genome sequences of *E. curvistellata* as a database (available at https://repository.lib.tottori-u.ac.jp/records/7586). The genes encoding the 38 kDa and 32 kDa proteins extracted from *V. pourtalesii* spicules were also identified by blastp of ncbi-blast-2.12.0⁺ program using the amino-terminal sequences of these proteins as a query against the whole translated transcriptome of *V. pourtalesii*[15] (see Methods section "Differential expression of genes" for details on *V. pourtalesii* transcriptome).

The 3D secondary structure for proteins of interest was inferred using two different software approaches: SWISS-MODEL[83,84] and Alphafold2[85] implemented in ColabFold - v1.5.2 platform[86] and using the MMseqs2 homology search algorithm. The highest scoring 3D protein structures were downloaded in "pdb" format and examined using the software Geneious Prime 2022.0.1 (https://www.geneious.com).

Occurrence of signal peptides in the sequences of extracted proteins was analyzed with Phobius web server (https://phobius.sbc.su.se/), under the "normal prediction" option.

## Protein location by immunohistochemistry

To locate hexaxilin, perisilin and glassin within the spicules rabbit polyclonal antibodies were prepared by Sigma-Aldrich Japan (Tokyo, Japan). Primary antibodies against glassin (1: 1,000 dilution) and hexaxilin (1: 1,000 dilution) of *E. curvistellata* were used in serum form. Antibodies against hexaxilin and perisilin of *V. pourtalesii* were affinity-purified (1: 100 dilution). Randomly fractured spicules from two individuals per species were incubated in 4% formalin in saline phosphate buffer (150 mM NaCl, 10 mM sodium phosphate, pH 7.5, PBS) for 30 min, then rinsed with PBS three times. Three batches of fixed spicules were incubated in 1.5% normal goat serum in PBS for 1 h, then in both anti-peptide (i.e., for hexaxilin 1, glassin, perisilin, etc.) and peptide rabbit antisera, which were 1000-times diluted with the blocking solution for 1 h. Pre-immunized rabbit sera were used as negative controls instead of the antisera. After PBS rinsing three times, spicules were Alexa 488 conjugated to goat anti-rabbit IgG antibody (Thermo Fisher Scientific, Waltham, MA, USA) 1000-times diluted with the blocking solution for 1 h. After three new rounds of PBS rising, spicules were mounted in Prolong Diamond antifade mountant (Thermo Fisher Scientific) and studied under a laser confocal scanning fluorescent microscope Olympus Fluoview FV10i (Tokyo, Japan). When necessary, 3D images were built by z-stack reconstructions of sequentially acquired images.

## Molecular phylogenetic analysis of proteins from silica

Searches of sequence similarity for proteins extracted from the silica skeletons of *E. curvistellata* and *V. pourtalesii* were performed by blast against both local and online databases. Local blast was performed using the program ncbi-blast-2.12.0⁺ against 36 open-source genomes and transcriptomes, which reasonably represented the main Porifera lineages, the main lineages of the rest of metazoans, and two (silicified vs non-silicified) choanoflagellates (Supplementary Data 1, 5, 7). Choanozoa and Porifera data used in local blast included: *Euplectella curvistellata* genome[30]; transcriptomes or genomes of *Vazella pourtalesii*[15], *Acanthoeca spectabilis*, *Hyalonema populiferum*, *Kirkpatrickia variolosa*, *Latrunculia apicalis*, *Periphylla periphylla*, *Rossella fibulata*, *Salpingoeca pyxidium*, *Sympagella nux* (data deposited to figshare https://doi.org/10.6084/m9.figshare.1334306.v3)[87]; *Aphrocallistes vastus* transcriptome (data deposited to University of Alberta Libraries Education and Research Archive, https://doi.org/10.7939/R3S000)[88] and the genome(https://zenodo.org/record/7970685)[31]; transcriptomes or genomes of *Haliclona amboinensis*, *Haliclona tubifera*, *Xestospongia testudinaria*, *Ephydatia muelleri*, *Stylissa carteri*, *Oscarella carmela*, *Leucosolenia complicata*, *Sycon ciliatum* (http://www.compagen.org/datasets.html); *Petrosia ficiformis*, *Crella elegans* (https://dataverse.harvard.edu/dataset.xhtml?persistentId=doi:10.7910/DVN/25071); transcriptomes of *Pseudospongosorites suberitoides*, *Chondrilla nucula*, *Ircinia fasciculata*, *Corticium candelabrum* (https://dataverse.harvard.edu/dataset.xhtml?persistentId=doi:10.

7910/DVN/24737), and *Oscarella lobularis* (https://www.ncbi.nlm.nih.gov/assembly/GCA_947507565.1). Blast online was performed at the NCBI site (https://blast.ncbi.nlm.nih.gov/Blast.cgi?PROGRAM=blastp&PAGE_TYPE=BlastSearch&LINK_LOC=blasthome) using the blastp option and at the EukProt v3 site (https://evocellbio.com/eukprot/) using the blast server option against the entire dataset, which includes genomes of 993 eukaryote species[89]. For both local and online blast, only hits showing a bit-score higher than 50 were retained, as it has been shown as a reasonable threshold criterion for inferring homology[90]. Protein isoforms were identified and distinguished from related protein sequences whenever possible, following the isoform prediction criterion implemented in Trinity-v2.11.0. To search for distantly related homologous proteins (Supplementary Data 2, 6), we used the Foldseek web server[91] (https://search.foldseek.com/search), which, differing from blast, performs the homology search based on the 3D structure of proteins rather than on sequence similarity. We set the AlphaFold/UniProt50 v4 as database, the 3Di/AA as mode and used the 3D protein structures previously inferred with Alphafold2 as input (for details on inference of protein structure, see section Extraction and identification of proteins from sponge silica).

The aa sequences of each de novo-characterized protein (accession numbers in Supplementary Data 8) and their putative homologs were aligned using the MAFFT (version 7) online service with the E-INS-I strategy[92]. Phylogenetic trees were estimated using both Maximum Likelihood and Bayesian Inference methods, implemented in the programs IQ-TREE 1.6.0[93] and MrBayes 3.2[94], respectively. In IQ-TREE 1.6.0, phylogenetic inference was performed using 5.000 replicates of ultrafast bootstrap[95] and enforcing ModelFinder Plus option, which both searches and applies the best fitting evolutionary model to the data[96]. The selected model for hexaxilins was the Le-Gascuel with a five-category FreeRate model for heterogeneity across sites (LG + R5). For perisilins, Whelan and Goldman plus a four-category FreeRate model (WAG + R6) was used. For glassins, we run the revised JTT matrix with empirical AA frequencies plus a four-category discrete Gamma Distribution (JTTDCMut+F + G4). The phylogenies inferred with MrBayes 3.2 were performed with two runs of 2.000.000 generations, four chains, sampling every 200 generations and applying the default burn-in of 25%. For each protein, the best fitting evolutionary model among the ones available in MrBayes 3.2 was selected through ModelFinder implemented in IQ-TREE 1.6.0. The selected model for both hexaxilins and perisilins was Whelan and Goldman with invariable sites plus a four-category discrete Gamma Distribution (WAG + I + G4), while Whelan and Goldman plus a four-category discrete Gamma Distribution (WAG + G4) was run for glassins. In the case of perisilins, the C- and N- terminus regions of the alignment contained high percentages of missing data, which led to low node support and inconclusive polytomies in the ingroup and outgroup. Thus, to improve the resolution of the phylogenetic relationship in the data matrix, those regions were excluded. Resulting trees were visualized with FigTree Software (http://tree.bio.ed.ac.uk/software/figtree/) and prepared for final figures with Adobe Illustrator 2023 (version 27.1.1).

### Differential gene expression of proteins for silica

To assess how ambient dSi availability affects the expression of genes encoding for the silicifying proteins, we used transcriptomic data from a dSi-enrichment experiment conducted with twelve individuals of *V. pourtalesii* and published elsewhere[15]. In brief, the transcriptomes were obtained for six individuals exposed to naturally low dSi concentration (15.6 ± 0.7 μM) and six individuals exposed to an ambient dSi concentration that was progressively increased from 10 μM to 200 μM over the course of two weeks. Measurements of dSi uptake indicated that the sponges subject to the dSi enrichment increased their dSi uptake rates relative to the other group (control) and, thus, their silicification activity[15]. Those experimental individual transcriptomes remained an

unparalleled database to examine the expression of the silicifying and silica-related proteins concerned in this study: hexaxilins, perisilins, glassins, and actin. For the current study, raw reads were downloaded from the Short Read Archive (SRA) under the BioProject number PRJNA580361[15]. Reads were filtered for quality with Trimmomatic version 0.39[97] (using parameter options as detailed elsewhere[15]) and assembled de novo with Trinity-v2.11.0[98,99] (assembly available at Figshare). Transcript abundances for each individual were then quantified using Kallisto[100], as implemented in Trinity-v2.11.0, and normalized. The Bioconductor package edgeR[101] (R version 4.1.0), supported by Trinity-v2.11.0, was used subsequently to examine whether transcripts of hexaxilin, perisilin, glassin, and actin were upregulated with statistical significance in individuals exposed to the high dSi treatment relative to the group of control individuals. Differential expression was determined using the quasi-likelihood (QL) F-test implemented in the statistical function GLMTreat (which does not require normally-distributed response data and reflects the uncertainty in estimating the dispersion of response for each gene) and relative to a four fold-change threshold of expression[102]. Therefore, only transcripts with a corrected P-value of false discovery rate (FDR) lower than 0.05 and at least with a fourfold expression level were considered as differentially expressed genes.

### Transmission electron microscopy

Transmission electron microscopy is a technically challenging approach for hexactinellid sponges because their large amounts of silica pieces prevent obtaining suitable ultrathin sections, unless the tissue is previously desilicified in hydrofluoric acid (HF), which eliminates spicules, leaving just empty holes at their location and obscuring biological interpretation. Besides, while skeleton removal makes ultra-microtome cutting feasible, it converts the tissue into a flaccid, mucus-like material where internal spaces collapse, making the original 3-D structure unrecognizable. Tissue pieces of about 2 mm³ were dissected from 3 individuals subjected to the experimental high dSi treatment. In order to maximize the chances of finding events of cellular silicification in the tissue, we selected the three individuals showing the highest rates of dSi consumption. Tissue pieces were immersed for 3 h in a fixative cocktail consisting of 2% glutaraldehyde, 2% osmium tetroxide, 65% sodium acetate buffer, 11% sucrose, and 20% distilled water[103]. After rinsing in distilled water, an initial, "field" dehydration step took place in 50% (3 min), before preserving samplings in 70% ethanol for 1 month, until their arrival to the laboratory for further processing. Then, some of these samples were rehydrated in distilled water and desilicified in 5% HF for 5 h prior to dehydration; the rest of the samples were subjected to dehydration without desilicification. Dehydration was then resumed in 70% (10 min), 80% (10 min), 90% (3 × 10 min), 96% (3 × 10 min) and 100% ethanol (3 × 10 min), followed by propylene oxide (2 × 10 min). Embedding in Spur resin required five immersion steps with gentle shaking during each one: 6 h in a 3:1 propylene-oxide/resin solution, 12 h in 2:2 propylene-oxide/resin solution, 7 h in a 1:3 propylene-oxide/resin solution and two 6-h steps in pure resin. The resin was hardened at 60 °C for 2 days. Ultrathin sections were obtained with an Ultracut Reichert-Jung ultramicrotome, mounted on gold grids and stained with 2% uranyl acetate for 30 min, then with lead citrate for 10 min. Observations were conducted with a JEOL 1010 TEM operating at 80 kv and provided with an external Gatan module for the acquisition of digital images. No statistical analyses were conducted on the ultrastructural findings.

### Statistics and reproducibility

All individuals used in the study were randomly collected and randomly assigned to treatments, unless otherwise is specified in the pertinent Method section. No statistical method was used to pre-determine sample size for the different experiments; we used *n* = 6

(individuals) for each of the two groups in the differential gene expression study and $n = 3$ (spicule or tissue samples) for the immunostaining and transmission electron microscopy observations. Statistical analyses were not conducted on immunostaining and electron microscopy findings. Western blot gels were not interpreted through quantitative or semiquantitative comparisons. No animals or data were excluded from the analyses. Comparisons for determining differential gene expression were based on a quasi-likelihood $F$ test, as implemented in the function GLMTreat of EdgeR through Trinity-v2.11.0. The level of significance of tests ($P > 0.05$, $P < 0.05$, $P < 0.01$, $P < 0.001$) is provided in the figures, while corrected exact $P$ values are provided in Supplementary Information. Full transcriptome and full genome data were searched with no previous filtering. Researchers were not blinded to allocation during experiments and outcome assessment.

### Reporting summary

Further information on research design is available in the Nature Portfolio Reporting Summary linked to this article.

## Data availability

All sequence data generated in the present study, including the newly assembled transcriptome of *Vazella pourtalesii*, have been deposited in Figshare data repository (https://doi.org/10.6084/m9.figshare.23799351). Protein sequences are available at GenBank database, with accession codes and hyperlinks provided in Supplementary Data 8. Additionally, protein sequences are also available at the DNA Data Bank of Japan (https://www.ddbj.nig.ac.jp/index-e.html). The sources of all previously published genomes and transcriptomes used for local blast searches are available as hyperlinks in Supplementary Data 1, 5, and 7. The unpublished genome of *Euplectella curvistellata* herein used is available at Tottori University repository (https://repository.lib.tottori-u.ac.jp/records/7586). Raw transcriptomic reads of the twelve individuals of *Vazella pourtalesii* from a dSi-enrichment experiment published elsewhere[15] were downloaded from Short Read Archive (SRA) under the BioProject number PRJNA580361. All quantitative and qualitative raw data supporting the findings and used for the statistical analyses are given in one Supplementary Table, eight files of Supplementary Data and seven files of Source Data provided with this paper. Source data are provided with this paper.

## Code availability

No specific code was written or developed for this study. We used regular scripts of Trinity software (https://github.com/trinityrnaseq/trinityrnaseq/wiki), following pipelines detailed in Methods.

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

## Acknowledgements

The authors thank I. G-Martos for help with spicule drawings, A. Riesgo for advice when designing and developing the transcriptomic pipeline, G. Muricy by comments on Homoscleromorpha, and J. M. Elías for help

on in-house script development. This work was supported by a Ministry of Education, Culture, Sports, Science and Technology Grants-in-Aid for Scientific Research of Japan (Grant 15K06581, 19K05164) to K.S., J.A. and T.B, and by a H2020 grant (RIA-679849-2) and a Ministry of Science, Innovation and Universities of Spain grant (PID2019-108627TRB-100) to M.M.

## Author contributions

K.S., M.M., L.L. and J.A. conceived the study and designed experiments and analyses. K.S., M.N., Y.S., H.K., T.B. and J.A. extracted, sequenced, and immune-located proteins. M.M. conducted electron microscopy. L.L., M.M. and K.S. performed phylogenetic and gene expression analyses. M.M., K.S. and L.L. assembled the manuscript with input from all authors.

## Competing interests

The authors declare no competing interests.
