## [Peer Review File · Nature Communications]

Silica-associated proteins from hexactinellid sponges support an alternative evolutionary scenario for biomineralization in PoriferaEditorial Note: Parts of this Peer Review File have been redacted as indicated to maintain the confidentiality of unpublished data.

REVIEWER COMMENTS

Reviewer #1 (Remarks to the Author):

The manuscript entitled "Novel silicifying proteins from glass sponges overturn the evolutionary scenario for biomineralization in Porifera and relate to silicification in other metazoans" is an experimental study on two novel proteins, hexaxilin and perisilin, involved in the silicification process in two sponge species, *Euplectella curvistellata* and *Vazella pourtalesii*. Biomineralized parts were extracted according to established procedures which yielded another novel sponge silica protein, glassin, by the same research groups few years ago. The authors used straight-forward methodologies for characterization of the protein extraction and fractionation. Based on the obtained novel sequence data and other sources (sponges genomes and transcriptomic data), the authors performed an extensive molecular phylogeny analysis by means of several selected bioinformatics resources in order to reveal evolutionary relationships between major sponge groups and other silicifying phyla. The obtained results were revisited against the state-of-the-art in the field for developing a hypothesis for a most likely scenario of biosilica formation under biomolecular control. Immunohistochemistry experiments provide strong indications for the co-localization with and within silica deposits. Perhaps, the results and conclusions of differential gene expression results could be reported and summarized more clearly. Despite some shortcomings in terms of silica preservation, ultrastructural TEM data are also included for illustration. In summary, all results presented here are noteworthy because the interplay between some key proteins seems to trigger function and evolutionary relationships, in contrast to previous reports.

Perhaps one of the most appreciated key features of this paper is the unusually critical evaluation of the new data against previous reports on how genetically controlled silicification actually works which may even lead to some conceptual paradigm shift for sponge spicule formation and mineralization. Without any doubt, this work will be of significance to the field and related fields (even broader relevance!). Conclusions and claims are well supported by the data, and - if any - methodological shortcomings have been considered with great care. I could not detect any flaws in the data analysis, interpretation and conclusions, which would prohibit publication or require revision. The methodology is sound and meets the expected standards in the field. The paper is very well written and it is assumed that the published version will hopefully provide more details on how the sequences (or fragments) can be accessed directly in the databases, which will certainly motivate further fundamental and applied research initiatives.

Reviewer #2 (Remarks to the Author):

Based on the discovery of lineage-specific proteins recovered from hexactinellid sponge spicules, this study argues that spicules have evolved multiple times independently in different sponge lineages. This is an interesting result with significance for understanding evolutionary biology, and biomineralization more generally. Below are my comments and questions for the authors:

1. I question whether the proteomic methods applied are the most appropriate. My understanding is that the authors isolated spicules and dissolved them, then ran the insoluble fraction of the purified protein on an SDS-PAGE gel, extracted 38, 23, and 32 kDa bands for sequencing by Edman-degradation. While this strategy seems to have worked to

identify the bands of interest, it is limiting. Why analyze all recovered proteins by LC/MS/MS to get a comprehensive look at the spicule-associated protein diversity? As you discuss (and I will address below) there are reports of other proteins (such as actin) in spicules and this would be an opportunity to independently evaluate those claims.

2. I'm a bit skeptical of the limited extent of antibody validation performed. As reported, the authors produced polyclonal antibodies against short synthetic peptide epitopes, and then performed immunostaining relative to pre-immune serum. The antibodies (not affinity purified?) were reportedly used at 1:1000 dilution (a very high dilution, what was the starting concentration of each? And were they all the same?) without titration or additional validation. Minimally it would seem prudent to perform a Western blot on the spicule "lysates" to validate that these antibodies recognize the appropriate band. Ideally, these antibodies could be used to perform immunoprecipitation from spicule lysates, and the precipitate analyzed by LC/MS/MS.

3. The hexalin antibody staining clearly shows axial filament staining, but there does seem to be some faint staining elsewhere in the spicule. It would be interesting to see these images at higher exposure levels. Also, given that these are crude antisera and with limited validation, it would be more convincing to see multiple examples of each staining pattern, for each antibody.

4. The bioinformatic methods essentially involved using BLAST searches to identify possible homologs, then aligning similar proteins and building phylogenetic trees. One suggestion for the authors is to consider also searching a more inclusive database than NCBI. Notably, EukProt (<https://evocellbio.com/eukprot/>) is much more comprehensive and may help them discover distant homologs in protists not represented in NCBI. Also, I would encourage the authors to at least check their SwissModel structural predictions by comparison to AlphaFold2. It may also be valuable to use Foldseek to identify structurally similar proteins in distant lineages, rather than just BLAST search.

5. Do perisilin and glassin have signal peptides consistent with their secretion into the extracellular space? This seems like an important consideration to support the argument that they are involved in extracellular silica deposition and could be analyzed using software like Phobius. If not, this should be explicitly discussed.

6. Why have the authors excluded hexalin-like sequences from non-sponge animals from their phylogenetic analysis? It seems important to include these to see the effect on the topology, particularly in light of arguments that the hexalin-like sequences in lineages like Brachiopods may also be involved in silica deposition.

7. As written, it seems that the authors accept the single published report of actin in spicules, but I think this should be viewed more skeptically at present. I think the report, in this study, of upregulated actin in response to elevated silica levels is more likely a secondary effect. For example, in demosponges it has been shown that actin-based cell junctions form at the cell-spicule interface. This, and countless other cellular processes, could contribute to elevated actin expression.

8. The authors include the following sentence, "Initial claims of silicatein in Hexactinellida are currently reinterpreted as either contamination or misinterpreted cathepsin-L enzymes involved in lysosomal digestion rather than silicatein." It is important to unpack this a bit and elaborate since the claim here is that demosponge spicules are unrelated to hexactinellid

spicules. It seems like it may even be necessary to include some supplementary analysis to refute this claim if the authors don't trust it.

9. I disagree with the authors use of language such as higher animals to refer to non-sponge animals, and using terms like "ancient animals" to describe sponges. Sponges are one of many modern animal lineages, the first credible sponge fossils date back to the Cambrian along with other animals. I would refer the authors to Dunn, Leys, Haddock "The hidden biology of sponges and ctenophores" for arguments in favor of this perspective.

10. I conclude with a suggestion (certainly not a mandatory revision). This study is interesting and convincing in its own right, but it would be considerably strengthened if the authors were to express recombinant forms of these proteins and show that they nucleate silica deposition in vitro. As is, I think their interpretation of these proteins as enzymes involved in spicule formation is likely right, but experimental evidence would make the argument much stronger.

Reviewer #3 (Remarks to the Author):

Review NCOMMS-23-13837

Novel silicifying proteins from glass sponges overturn the evolutionary scenario for biomineralization in Porifera and relate to silicification in other metazoans by Shimizu et al.

In this manuscript, Shimizu and colleagues investigate some of the proteinaceous components of the spicules of Hexactinellid sponges. They discover two new silicifying proteins (they call them hexaxilin and perisilin) and furthermore another protein that was discovered earlier, glassin.

While this is an interesting study there are a number of issues that prevent publication at the present stage. I will start with the important broader issues and then list more detailed ones.

1) and this is very important: the authors base their work and conclusions on a genome that apparently have sequenced, of the species *Euplectella curvistella*. They cite two previous papers for this genome

29. Shimizu, K. et al. Exploration of genes associated with sponge silicon biomineralization in the whole genome sequence of the hexactinellid *Euplectella curvistellata*. in *Biomineralization* (eds. Endo, K., Kogure, T. & Nagasawa, H.) 147–153 (Springer, 2018). doi:10.1007/978-981-13-1002-7_16.

30. Shimizu, K. et al. Glassin, a histidine-rich protein from the siliceous skeletal system of the marine sponge *Euplectella*, directs silica polycondensation. *Proc. Natl. Acad. Sci. USA* 112, 11449–11454 (2015).

However, the genome (which, according to Reference 29 has a questionable quality with an N50 of 420 bases! No wonder why glassin could not be retrieved from this "genome" in full (page 9, line 7) ...

See Table 16.1 from citation 29.

In addition, this genome has not been made publicly available yet, I checked GenBank and no genome assembly is available there.

It is my foremost request that the authors MUST make all the data of their study, including the full genome and transcriptome data of the species investigated, as well as all MS raw files and code for analyses publicly available, for example in a GitHub repository, or on Data Dryad or Figshare. Without all underlying data being used made available according to the FAIR principles, the study cannot be repeated and the validity assessed.

In addition, providing sequences and their alignments only as Figures is useless; please provide all of them in a usable format, such as Fasta or Nexus; a GitHub (or similar) repository would be most appropriate for this. For example, Supplementary Figs. 1, 4, 6 need to be provided as a useable alignment.

2) Ehrlich and colleagues published on the presence of several other components of (hexactinellid) sponge spicules, i.e., chitin (polysaccharide), collagen, and most recently actin. These findings are completely ignored in the Introduction, and barely mentioned/discussed in the Discussion.

I would like to see a more comprehensive treatment of the issue. Especially the puzzling occurrence of actin in the glass sponge spicules studied by Ehrlich et al. (DOI: 10.1002/adv.202105059) needs consideration; actin had a molecular weight of 42 kDa, but I do not see a band here in Fig. 2 at that size – did it slip through the sample preparation? A prominent target for investigation would be the axial filament, which Shimizu et al. claim to be Hexaxilin, and Ehrlich et al. to be actin.

So, there is some discrepancy here that needs to be resolved, for example by co-staining with antibodies – Ehrlich et al. used a primary anti- β -actin antibody and secondary (anti-rabbit IgG (H+L), F(ab')₂ Fragment (Alexa Fluor 488 Conjugate) antibodies, so these analyses should be repeated with your species.

One problem here might be the differences in the sample preparation: in the present study the authors used HF treatment to dissolve the silica spicules, in some of Ehrlich et al.'s work for the detection of collagen and chitin they used NaOH, but for the detection of actin also HF.

So, I suggest, for better comparability, that the authors also prepare the organic fraction of the spicules with NaOH instead of HF and analyse the protein fraction remaining. I find it quite puzzling that Shimizu et al. did not detect any collagen, chitin, or especially actin (which was detected by Ehrlich et al. 2022 by also dissolving the spicules with HF). This really needs detailed addressing.

As it stands now, the reader is really left confused with what is going on with silification in hexactinellid sponges; I think the authors need to bring all this together.

3) While the text reads in large parts OK, there are sections where the writing is difficult to follow; one example is the abstract, which needs a thorough proofread, as does the whole manuscript.

More detailed comments and suggestions:

ABSTRACT:

As mentioned above, needs complete overhaul in terms of language.

L20: Sponges are not the oldest animal lineage; they are as old as their sister group.

L23: “confirm” is a very strong word, I do not see (esp. due to a lack of in-depth characterization and consideration of the Homoscleromorpha) such a confirmation. Should be softened to “suggest”.

L31: There are no “higher” metazoans. This is incorrect phylogenetic terminology that puts vertebrates / humans at the top – we are past those days. Why not be more specific and replace with “bilaterians”?

INTRODUCTION:

Some language improvements needed.

P2 L2: consider adding “extant” to Porifera, because, according to Systema Porifera, there is another extinct class of the Porifera, the Archaeocythida.

P3 L11/12: unclear to me what you mean with “dissolved condition”. Consider to explain this better.

P3 L14: Here you should provide more information about other recent findings of organic components isolated from Hexactinellid spicules, such as chitin, collagen, and actin – see Herman Ehrlich’s work (and also see comment above).

P3 L22: Change “the findings provide novel mechanistic insights” to “the findings suggest novel mechanistic insights”. After all, without functional experiments you cannot really prove function (the mere presence of a protein does not necessarily prove its function, it just suggests involvement).

P3 L24: replace “higher metazoans” (see comment above) with “bilaterians”

RESULTS & DISCUSSION

P3 L34: Where is the genome accessible at? Without the genome publicly accessible and assessable the study cannot be replicated and the findings validated. As mentioned above, making the genome public is a non-negotiable requirement for publication of this study.

P3 L29 – P4 L20: I am puzzled that the authors did not detect at all in their spicules any other biological components that were suggested to be present in the Hexactinellida, such as chitin, collagen, and action (see comments above). If so (no traces) this should explicitly be stated here somewhere in this section.

P4 – P10: The subsections on Hexaxilin, Persilin, and Glassin are very long, wordy, and in parts difficult to read/follow. Consider shortening and/transferring parts into the Supplementary Material.

P5 L15: “is acceptable for Porifera”. Unclear and not well substantiated statement. What is this based on, how is this judged, where is the threshold? Do you have a reference for this? Otherwise, rephrase please.

P6 L5: I am not really convinced by your reasoning about the origin of the MBLf gene through horizontal gene transfer. I think a more in-depth analysis of this gene in both the Hexactinellida and the bacteria is in order for a better substantiation of this claim. Have other cases of horizontal gene transfer been reported and substantiated in the Porifera?

P7 L7 “Because homosclerophorid sponges lack silicatein and hexaxilin, ...” Do they? Did you analyse any homosclerophorid sponges with the same methods? Or do you have a reference for this statement where this has been done?

P7 L16: “of the ancestral ocean where the major sponge classes arose” (citation 15). Please provide more details here about when this happened, and what the likely ocean conditions

were at that time. I suggest to cite a recent molecular clock study here that specifically deals with the timing of the divergences of the animal lineages.

P8 L9-11: This sentence is not well phrased and is largely speculation. It needs more evidence.

P9 L7: again, make the genome publicly available; you refer here to a specific contig in the genome but that genome is not available.

P9 L30 – 36: this whole section reads quite confusing, please rephrase to make clearer. How does that impact your narrative if *Glassin* is absent from *V. pourtalesii*?

THE SILICIFICATION PROCESS IN HEXACTINELLIDA

P10 First paragraph of that section. The value of this section (and the associated TEM Figure 9) is elusive to me. How does that add to the interpretation of the data? In Fig. 9 no axial filament (or other proteinaceous components) is visible in the TEMs. To make such an ultrastructural study really useful to strengthen the case, you would need to carry out immunogold labelling to identify the location of the expression of, e.g., hexacilin. This would be of great value. Otherwise omit these TEMs, or shift them to the supplementary material.

P10 L19 (second paragraph): it would be really useful to provide a summary figure with a sketch of how you envisage the silification process in the Hexactinellida, and what is different to the Demospongiae.

P10 last paragraph of that page and the first paragraph of P11: you treat the issue of the other “non-silicifying” proteins (like collagen and actin) very superficially, without much depth. You will need to provide evidence (see also comment above) that the two proteins are not present in the species you studied, and if you indeed detect them, how they interact with the proteins you discovered. Either by antibody labelling, as suggested above, or by in situ hybridization. Without this, your study is incomplete.

P11 L12: the diatoms appear quite sudden and random. Needs better integration.

CONCLUSIONS

Need complete overhaul, some of the statements there are too strong and insufficiently substantiated.

Figure 4:

Please provide species names in the tree at the leaves and no abbreviations (this applies to ALL trees!)

The class is Demospongiae not Demospongia (correct in all trees)

Generally: please provide more information about the hexacilin discovered in the Calcarea; at present the explanation in the text is insufficient. Provide a deeper analysis of the genes discovered, what are the exact differences to the gene in Hexactinellida and Demospongiae, where they apparently are involved in silicification. Same applies to Persilin.

EXPLANATIONS TO REVIEWERS

Reviewer #1 (Remarks to the Author):

The manuscript entitled "Novel silicifying proteins from glass sponges overturn the evolutionary scenario for biomineralization in Porifera and relate to silicification in other metazoans" is an experimental study on two novel proteins, hexaxilin and perisilin, involved in the silicification process in two sponge species, *Euplectella curvstellata* and *Vazella pourtalesii*. Biomineralized parts were extracted according to established procedures which yielded another novel sponge silica protein, glassin, by the same research groups few years ago. The authors used straightforward methodologies for characterization of the protein extraction and fractionation. Based on the obtained novel sequence data and other sources (sponges genomes and transcriptomic data), the authors performed an extensive molecular phylogeny analysis by means of several selected bioinformatics resources in order to reveal evolutionary relationships between major sponge groups and other silicifying phyla. The obtained results were revisited against the state-of-the-art in the field for developing a hypothesis for a most likely scenario of biosilica formation under biomolecular control. Immunohistochemistry experiments provide strong indications for the colocalization with and within silica deposits. Perhaps, the results and conclusions of differential gene expression results could be reported and summarized more clearly. Despite some shortcomings in terms of silica preservation, ultrastructural TEM data are also included for illustration. In summary, all results presented here are noteworthy because the interplay between some key proteins seems to trigger function and evolutionary relationships, in contrast to previous reports.

Perhaps one of the most appreciated key features of this paper is the unusually critical evaluation of the new data against previous reports on how genetically controlled silicification actually works which may even lead to some conceptual paradigm shift for sponge spicule formation and mineralization. Without any doubt, this work will be of significance to the field and related fields (even broader relevance!). Conclusions and claims are well supported by the data, and - if any - methodological shortcomings have been considered with great care. I could not detect any flaws in the data analysis, interpretation and conclusions, which would prohibit publication or require revision. The methodology is sound and meets the expected standards in the field. The paper is very well written and it is assumed that the published version will hopefully provide more details on how the sequences (or fragments) can be accessed directly in the databases, which will certainly motivate further fundamental and applied research initiatives. **Author Response (AR):** Thank you for the kind words about our study and its significance. We have revised the explanations on differential gene expression, attempting to streamline them.

Reviewer #2 (Remarks to the Author):

Based on the discovery of lineage-specific proteins recovered from hexactinellid sponge spicules, this study argues that spicules have evolved multiple times independently in different sponge lineages. This is an interesting result with significance for understanding evolutionary biology, and biomineralization more generally.

AR: We thank the reviewer for recognizing the significance of our study.

Below are my comments and questions for the authors:

1. I question whether the proteomic methods applied are the most appropriate. My understanding is that the authors isolated spicules and dissolved them, then ran the insoluble fraction of the purified protein on an SDS-PAGE gel, extracted 38, 23, and 32 kDa bands for sequencing by Edman-degradation. While this strategy seems to have worked to identify the bands of interest, it is limiting. Why analyze all recovered proteins by LC/MS/MS to get a comprehensive look at the

spicule-associated protein diversity? As you discuss (and I will address below) there are reports of other proteins (such as actin) in spicules and this would be an opportunity to independently evaluate those claims.

AR:

Regarding, methodological appropriateness, please, note that our approach to extract and identify proteins from the sponge silica has provided nearly all the scientific knowledge that is currently available regarding silicifying proteins in the phylum Porifera. It first allowed the discovery of silicatein (in 1999), then of glassin (in 2015) and, in this manuscript, of hexaxilin and perisilin. Furthermore, the approach has recently revealed a fifth new protein in the class Homoscleromorpha **[Redacted]**. Thus, our methodological approach for extracting silicifying proteins from the sponge silica works well and it would have not been a good idea to modify it just to attempt detecting (or not detecting) actin, which is not a silicifying protein but a protein accompanying the silicifying proteins of the axial filament and that goes beyond the scope of our study. Please, note that our SDS-PAGE and Western blot analyses of silica extracts revealed well-marked bands of glassin, perisilin and hexaxilin, as corresponding to major protein components of the studied spicules (See Response Figure 2 at the bottom of this letter, which will become new Supplementary Fig. 1 in the revised manuscript). Additionally, we found some other proteins in the gels, all making faint bands of molecular weight higher than 70 kDa (see red arrows in Response Figure 2). As now explained in the revised manuscript, we could not obtain sequence information from those minority bands, because protein abundances in the gels were systematically low. Since actin is a 42 kDa protein, it can be ruled out that any of those minority protein bands of high molecular weight could be actin.

[Redacted]

Response Figure 2= new Supplementary Fig. 1. SDS-PAGE and Western blot (Wb) analysis of extracts from hexactinellid silica. SDS-PAGE analysis (a, c) of the combined water-soluble and water-insoluble spicule extracts of *Euplectella curvistellata*, along with Wb analyses of that combined extract with 1:1000 anti-hexaxilin antiserum (b) and the soluble fraction of the extract with 1:1000 anti-glassin antiserum (d). Note that the monomeric units of these proteins appear able to self polymerize easily, forming dimers and trimers retained in faint bands (white arrow heads) of high molecular weight (MW), which are also specifically labelled by the corresponding antibodies. SDS-PAGE analysis of the water-soluble and water-insoluble spicule extracts of *Vazella pourtalesii* (e), along with Wb analyses of the insoluble extract with 1:100 anti-hexaxilin antiserum (f) and 1:100 anti-perisilin antiserum (g). Note, again, formation of faint bands of putative dimers and trimers labelled by the antibodies. Importantly, bands corresponding to actin (42 kDa) are not visible in any of the three SDS-PAGE analyses (a, c, e), while additional minority bands (red arrows) occur at molecular weights higher than 70 kDa.

As we have now explained in full detail in new Supplementary Text-3, our negative finding regarding actin does not mean that actin could actually occur. Why our approach aimed at revealing the silicifying proteins does not pick up the presence of actin in the hexactinellid

silica, unlike that reported by Ehrlich et al.¹ ? There are a number of potential reasons, including important differences between our methodological approach and that of Ehrlich et al.¹. First, we performed a more aggressive cleaning of the outer organic matter of the spicules by immersion in bleach and then in concentrated HNO₃/H₂SO₄ (1:4) at room temperature, whereas Ehrlich et al. immersed the spicules in only 70% HNO₃ at room temperature for 3 days. Our more aggressive cleaning makes it easier for bleach and/or acids to infiltrate between the loose silica layers characterizing many of the hexactinellid spicules² or entering through cracks in the silica that lead to the axial canal, posing in greater risk the integrity of some of the proteins of the axial filament. We also conducted a much more aggressive digestion of the silica by complete immersion of the acid-cleaned spicules in HF/NH₄F (4M:8M) for 2-3 days, whereas Ehrlich and co-workers¹ used only 10% HF (~5 M) droplets falling on the spicules for 7h to 10h. Another major difference in the approach is that we used the extracts from the total silica digestion, whereas Ehrlich and co-workers used the extracts from only axial filaments previously released from the silica. The process of concentration of the proteins in the extracts was also markedly different. Ehrlich et al.¹ were only able to retrieve actin (Ehrlich, Pers. Comm.) when the soluble extracts of the axial filament “were precipitated with ice cold acetone, incubated at -20°C and centrifuged at 10,000 g for 10 seconds” to obtain protein pellets. The need for such precipitation steps suggests that actin was in low concentration in those filaments. We did not perform such a precipitation step. For protein concentration in the soluble fraction, we used ultrafiltration. Theoretically, ultrafiltration would also concentrate actin, but, in practice, actin could also have been lost from our soluble fraction by selectively binding to the ultrafiltration membrane or by precipitating during ultrafiltration. There are also differences in the electrophoresis approach. We used NOVEX NuPAGE system, in which proteins always run at neutral pH conditions. In contrast, proteins are subjected to alkaline pH during electrophoresis in the conventional SDS-PAGE used by Ehrlich et al.¹. The NuPAGE system gives better resolution than the conventional system and, in fact, glassin does not appear in the conventional SDS-PAGE system. In this regard, we do not know whether actin might also behave unusually in the NuPAGE system. In summary, due to the aforementioned methodological differences, the absence of actin in our gels cannot be used to decide whether this protein is actually absent or present in the silica of *E. curvistellata* and *V. pourtalesii*.

In summary, the revised version addresses in more detail the issue of actin. The reviewer should also understand that the primary objective of our study is to unravel the evolution of silicifying proteins and, wherever possible, their action mechanism for silica polycondensation. Actin is not a silicifying protein. According to Ehrlich and co-workers, actin would act as a template around which the units of the actual silicifying proteins (i.e., silicatein) would assemble to form the axial filament. Such a role would not be exclusive to spicules of Hexactinellida, since actin is reported from both tissue and spicules of all three siliceous sponge classes. Because of such widespread occurrence in Porifera and because actin has not active participation in silica polymerization, actin was (and is) of little interest to our evolutionary objectives. Please, note that our study could never become a test for the presence/absence of actin in the silica of Porifera, as this reviewer would like. Dr. Ehrlich (in the 2022 paper and in the 11th World Sponge Conference) reported that actin occurs in all three classes of siliceous sponges. In our study, we are dealing with only Hexactinellida. Therefore, the suggested comprehensive protein analyses of the silica extracts would have to be conducted, in order to be relevant, not only in Hexactinellida, but also in Demospongiae and Homoscleromorpha. Of course, that would become a massive work needing a manuscript (and funding) of its own, and falling far beyond the scope of our current study, which is already quite complex. We hope the reviewer understands our situation.

2. I'm a bit skeptical of the limited extent of antibody validation performed. As reported, the authors produced polyclonal antibodies against short synthetic peptide epitopes, and then performed immunostaining relative to pre-immune serum. The antibodies (not affinity purified?) were reportedly used at 1:1000 dilution (a very high dilution, what was the starting concentration

of each? And were they all the same?) without titration or additional validation. Minimally it would seem prudent to perform a Western blot on the spicule “lysates” to validate that these antibodies recognize the appropriate band. Ideally, these antibodies could be used to perform immunoprecipitation from spicule lysates, and the precipitate analyzed by LC/MS/MS.

AR: Please, note that most of the reviewer’s concerns in this regard arise from assuming that we did not perform some antibody testing that we actually did do (although it is our fault not having explained it properly in the initial manuscript version). Please, see in Response Figures 2, 3, 4 that our immunostaining was strictly attached to the standards that this reviewer is requesting. First, we initially assayed the antisera on Western blot bands for specificity (Response Figure 2b, d, f, g) and corroborated that bands of hexaxilin, perisilin and glassin of *E. curvistellata* and *V. pourtalesii* were stained by their corresponding antibodies. Please, note also that these silicifying proteins appear able of self-polymerizing easily and that putative protein dimers and trimers spontaneously formed, which were also specifically labelled as additional bands of nearly double or triple molecular weight. Please, note also that, unlike assumed by the reviewers, ELISA titrations were conducted (Response Figure 3). They indicated that the antibodies were efficient

Response Figure 3. Results of ELISA titer for anti-hexaxilin serum of *Vazella pourtalesii*.

even at concentrations as low a 1:16,000. Thus, the antisera of *E. curvistellata* were not purified for further antibody enrichment, but the antisera of *V. pourtalesii* was affinity purified (1/100), as now explained in Methods. In the revised manuscript, we also inform that incubations were initially assayed at 1:100 and 1:1000. It was found that 1:100 was too concentrated (check antibody smearing in Response Figure 2g). In contrast, as it can be seen in Response Fig. 4 (= part of revised Fig. 2 in the main text), the 1:1000 concentration worked correctly. We apologize for not including all of these preliminary tests in the original manuscript, as we considered them obvious steps in the immunostaining process. In response to reviewer 2's comments, we provide all of these data in the revised manuscript. We hope that the reviewer is now more convinced that our immunostaining approach is correct and strictly adheres to standard protocols.

3. The hexaxilin antibody staining clearly shows axial filament staining, but there does seem to be some faint staining elsewhere in the spicule. It would be interesting to see these images at higher exposure levels. Also, given that these are crude antisera and with limited validation, it would be more convincing to see multiple examples of each staining pattern, for each antibody.

AR: Reviewer 2, despite admitting that our antibody strongly marks the hexaxilin axial filament, also enquired about the very faint extra-axial staining of the anti-hexaxilin. See Response Figure 4, for an explanation. Such a faint signal is residual staining resulting from defective rinsing after the antibody incubation. As you know, the sponge silica is glass, with the same general chemical formula than the glass of our windows. It means that, when spicules are fractured by us for the internal proteins to become exposed for immunostaining, nanometer-size microcavities are accidentally formed. Most of the antibody entering those tiny cavities during the incubation is normally washed out through the rinsing steps, but, in complex cavities, tiny traces of it may resist rinsing, producing the very faint staining of the silica noticed by the reviewer. Note that such faint signal has no spatial pattern other than being associated to areas of irregular fractures of the silica and we have indicated this in the revised figure (see red and yellow arrows in Response Figure 4). Note also that a very faint color signal also persists underneath of the outermost silica layer of the spicule of *Vazella*. The reason is the same, the outermost silica layer in *Vazella* spicules is quite loose (typically detaching) and the fluorochrome that enters underneath cannot be easily rinsed off (see Response Figure 5, extracted from Figure 4M in Maldonado et al., 2022)³. Altogether,

the reviewer should admit that those very faint signals have nothing to do with the strong mark produced by the axial filament or the presence of proteins in the peripheral silica.

Also, we mentioned in the previous response, we initially assayed the antibodies at two trial concentrations (1/100, 1/1000), the more diluted concentration produced the cleanest signal, because it was most efficiently rinsed from nanocavities in the silica after incubation. We have added this information to the revised manuscript.

4. The bioinformatic methods essentially involved using BLAST searches to identify possible homologs, then aligning similar proteins and building phylogenetic trees. One suggestion for the authors is to consider also searching a more inclusive database than NCBI. Notably, EukProt (<https://evocellbio.com/eukprot/>) is much more comprehensive and may help them discover distant homologs in protists not represented in NCBI. Also, I would encourage the authors to at least check their SwissModel structural predictions by comparison to Alphafold2. It may also be valuable to use Foldseek to identify structurally similar proteins in distant lineages, rather than just BLAST search.

AR: Thanks for these useful recommendations. After checking EukProt, no new hit for glassin was found in addition to previously recovered by us from other databases. Thus, glassin remains interpreted as a Metazoan (Hexactinellida) novelty. In contrast, some hexaxilin-like and perisilin-like hits were retrieved within Protista. We have conducted new phylogenies for these two proteins incorporating the new hits, in order to produce the most updated phylogenies for the revised manuscript (see below Response Figure 6 as an example, and revised Fig. 4 and 7 in the main manuscript). Interestingly, among protists, hexaxilin-like searches retrieved only 5 hits, with an unconventional character distribution across taxa, being all marine organisms. One is an Apicomplexa, the gregarine *Siedleckia nematoides*, which parasitizes polychaetes. The four remaining hits concentrate within the supergroup Rhizaria (Cercozoa + Foraminifera + Radiolaria), that is, protists emitting pseudopodia. However, the hexaxilin-like protein shows again a weird distribution across those rhizarian phyla. Three hits occur in Cercozoa (the chlorarachiphyte algae *Bigelowiella natans* and *Amorphoclora amoebiformis*, and the giant amoeba *Gromia sphaerica*). The fourth rhizarian is the foraminifer *Bolivina argentea*. Surprisingly, no other groups of protists contain a hexaxilin-like sequence homolog to that hypothetically inherited by metazoans from their protist ancestor. Also surprisingly, Radiolaria — the only silicifying rhizarians — lack hexaxilin-like sequences. In the case of perisilins, the EukProt search revealed six new hits belonging to five different Protist phyla (i.e., Amoebozoa, Ciliophora, Cryptista, Hemimastigophora and Foraminifera) and several additional metazoan hits not represented in NCBI.

Despite the new hits, the evolutionary significance of the silicifying proteins within Hexactinellida does not change. It is just that hexaxilin cannot be interpreted as the result of lateral gene transfer, nor perisilin can be interpreted as a metazoan novelty. Thus, we have reworded accordingly the pertinent text relative to those secondary aspects. All these information has been added to the revised manuscript.

Regarding the request to compare our SwissModel structural predictions with those of Alphafold2, we have done it (see new Supplementary Fig. 4-6, 9, 12 in the revised manuscript, and Response Figure 7 shown below as an example). We found no substantial differences in output, with Alphafold2 typically predicting the regions not available in the templates as random structures. We have included all this new information in our revised Results, Methods, and Supplementary Information (new Supplementary Figs 4-6, 9, 12).

We have also used Foldseek to identify structurally similar proteins in distant lineages, in addition to our previous BLAST searches. However, Foldseek searches did not retrieve hits in lineages different from those previously captured by BLAST at NCBI and EukProt. Anyway, we have added this information in the revised Results and Methods.

Response Figure 6 = revised **Fig. 4. Evolutionary relationships of hexaxilin.** Bayesian Inference (BI) phylogenetic tree of hexaxilins obtained with MrBayes 3.2. (Maximum Likelihood tree is shown in Supplementary Fig. 3). Included sequences correspond to all blast hits for hexaxilin-1 of *E. curvistellata* (sequence Ec_20423) with a bit score larger than 50 (Supplementary Table S1). Hexactinellid species are represented by names in different colour. Scale bar represents 0.4 amino acid substitutions per site. Posterior probability (pp) and bootstrap (b) values are given at each node as pp/b. Nodes with no bootstrap value were not recovered in the Maximum Likelihood phylogeny. Alignment data is available in fasta format in Source Data Fig. 4.

5. Do perisilin and glassin have signal peptides consistent with their secretion into the extracellular space? This seems like an important consideration to support the argument that they are involved in extracellular silica deposition and could be analyzed using software like Phobius. If not, this should be explicitly discussed.

AR: Thanks for this advice. We have processed sequences with Phobius and have now added to the revised manuscript that perisilin and glassin have signal peptides (1-18 aa) of conventional structure for being exported through the plasmalemma of the scleroblasts, in full agreement with our interpretation of extracellular functioning for these proteins. We indicate that hexaxilin also has a signal peptide (1-19 aa), which is consistent with its proposed translocation from the cytosol to the silica deposition vesicle for the formation of the axial filament necessary to initiate intracellular silica deposition. Thanks to this new analysis our proposal for action mechanism is now better supported. This information has been included in the revised Results and Methods, and in new Supplementary Table 7.

6. Why have the authors excluded hexalin-like sequences from non-sponge animals from their phylogenetic analysis? It seems important to include these to see the effect on the topology, particularly in light of arguments that the hexalin-like sequences in lineages like Brachiopods may also be involved in silica deposition.

AR: All sequences from non-sponge animals were included in the analyses and shown in the trees, but it seems that the reviewer overlooked them. Please, note in the phylogenetic tree that those 3 sequences were labelled at the phylum level. They all fell within the demosponge clade (see Fig. 4 in the original submitted version). In the revised manuscript, we now include a revised hexaxilin phylogeny (revised Fig. 4), which again includes non-sponge metazoans. Now, we have improved the labelling to prevent new mistakes by readers. Please, note also that some non-sponge animal clades are collapsed in the new revised figure for the sake of simplicity and readability (fully expanded clades can be found in the supplementary counterpart figure: Supplementary Fig. 4).

7. As written, it seems that the authors accept the single published report of actin in spicules, but I think this should be viewed more skeptically at present. I think the report, in this study, of upregulated actin in response to elevated silica levels is more likely a secondary effect. For example, in demosponges it has been shown that actin-based cell junctions form at the cell-spicule interface. This, and countless other cellular processes, could contribute to elevated actin expression.

AR: We have already addressed the issue of actin in our response to query number 1 and in our new Supplementary Text-3. We agree with the reviewer regarding the overexpression of actin. Thus, we have eliminated the comment from the main text and rephrased accordingly in Supplementary Text-3: “Interestingly, we identified three actin genes in the transcriptome of *V. pourtalesii*. One of them (*Actin-1*) has five isoforms, two of which (α and β) became drastically upregulated under high dSi concentrations (Supplementary Table S3-S4, Supplementary Fig. 13).

Again, our approach cannot decide whether such upregulation derives from an involvement of actin in the process of spicule production or it is resulting from the countless other cell processes in which this protein participates“.

8. The authors include the following sentence, “Initial claims of silicatein in Hexactinellida are currently reinterpreted as either contamination or misinterpreted cathepsin-L enzymes involved in lysosomal digestion rather than silicatein.” It is important to unpack this a bit and elaborate since the claim here is that demosponge spicules are unrelated to hexactinellid spicules. It seems like it may even be necessary to include some supplementary analysis to refute this claim if the authors don’t trust it.

AR: We did not express our idea correctly and have rephrased it. It is not that we are concluding in this study that hexactinellids do not have silicatein. This is not a new knowledge. It was already concluded in the review of silicateins by Riesgo et al. in 2015⁴. Since then, all subsequent transcriptomes and genomes of hexactinellid sponges have agreed that hexactinellids have no silicatein genes⁴⁻⁷. Likewise, recent revisions (published in June 2023) of the genome of the hexactinellid *Aphrocallistes vastus*⁸ and the genome of *Oopsacas minuta*⁹ have literally concluded that although earlier studies reported silicatein to also be present in Hexactinellida, no silicateins were subsequently found in glass sponge transcriptomes nor genomes. Those two very recent papers come in full agreement with the reported analyses for those two species in our initial manuscript version. Hence, there is overwhelming evidence that the only two early claims of silicateins in Hexactinellida likely represent mistakes. Also, note that silicatein has never been detected in our extractions of hexactinellid silica, despite the fact that part of this research team was the one who first discovered silicatein in sponge silica. In summary, there is overwhelming published evidence that hexactinellids lack silicatein and we have modified the text to include even the most recent works in this regards.

3. Riesgo, A., Maldonado, M., López-Legentil, S. & Giribet, G. A proposal for the evolution of cathepsin and silicatein in sponges. *J. Mol. Evol.* **80**, 278–291 (2015).
4. Shimizu, K. & Morse, D. E. Silicatein: A unique silica-synthesizing catalytic triad hydrolase from marine sponge skeletons and its multiple applications. in *Methods in Enzymology* vol. 605 429–455 (Academic Press Inc., 2018).
5. Shimizu, K. *et al.* Exploration of genes associated with sponge silicon biomineralization in the whole enome sequence of the hexactinellid *Euplectella curvistellata*. in *Biomineralization* (eds. Endo, K., Kogure, T. & Nagasawa, H.) 147–153 (Springer, 2018). doi:10.1007/978-981-13-1002-7_16.
6. Maldonado, M. *et al.* Cooperation between passive and active silicon transporters clarifies the ecophysiology and evolution of biosilicification in sponges. *Sci. Adv.* **6**, eaba9322 (2020).
7. Francis, W. R. *et al.* The genome of the reef-building glass sponge *Aphrocallistes vastus* provides insights into silica biomineralization. *R. Soc. Open Sci.* **10**, (2023).
8. Santini, S. *et al.* The compact genome of the sponge *Oopsacas minuta* (Hexactinellida) is lacking key metazoan core genes. *BMC Biol.* 2023 211 **21**, 1–21 (2023).

9. I disagree with the authors use of language such as higher animals to refer to non-sponge animals, and using terms like “ancient animals” to describe sponges. Sponges are one of many modern animal lineages, the first credible sponge fossils date back to the Cambrian along with other animals. I would refer the authors to Dunn, Leys, Haddock “The hidden biology of sponges and ctenophores” for arguments in favor of this perspective.

AR: OK. We have revised and refined our terminology through the manuscript to be more inclusive with all hypotheses on the origin of metazoans.

10. I conclude with a suggestion (certainly not a mandatory revision). This study is interesting and convincing in its own right, but it would be considerably strengthened if the authors were to express recombinant forms of these proteins and show that they nucleate silica deposition in vitro.

As is, I think their interpretation of these proteins as enzymes involved in spicule formation is likely right, but experimental evidence would make the argument much stronger.

AR: In the past, we run such an experiment successfully for glassin. Because of the potential biotechnological application of hexaxilin and perisilin recombinants, we have already assayed recombinants for their activity. However, the assayed sequences precipitated rapidly from the solution, preventing any further work. Though we keep working on it, we may need months/years before we can solve the problems of peptide stability in vitro. Thus, we are happy that your request is not mandatory, because the subject will need a separate study of its own.

Reviewer #3 (Remarks to the Author): Review NCOMMS-23-13837

Novel silicifying proteins from glass sponges overturn the evolutionary scenario for biomineralization in Porifera and relate to silicification in other metazoans by Shimizu et al.

In this manuscript, Shimizu and colleagues investigate some of the proteinaceous components of the spicules of Hexactinellid sponges. They discover two new silicifying proteins (they call them hexaxilin and perisilin) and furthermore another protein that was discovered earlier, glassin. While this is an interesting study there are a number of issues that prevent publication at the present stage. I will start with the important broader issues and then list more detailed ones.

1) and this is very important: the authors base their work and conclusions on a genome that apparently have sequenced, of the species *Euplectella curvistella*. They cite two previous papers for this genome

29. Shimizu, K. et al. Exploration of genes associated with sponge silicon biomineralization in the whole enome sequence of the hexactinellid *Euplectella curvistellata*. in *Biomineralization* (eds. Endo, K., Kogure, T. & Nagasawa, H.) 147–153 (Springer, 2018). doi:10.1007/978-981-13-1002-7_16.

30. Shimizu, K. et al. Glassin, a histidine-rich protein from the siliceous skeletal system of the marine sponge *Euplectella*, directs silica polycondensation. *Proc. Natl. Acad. Sci. USA* 112, 11449–11454 (2015).

However, the genome (which, according to Reference 29 has a questionable quality with an N50 of 420 bases! No wonder why glassin could not be retrieved from this “genome” in full (page 9, line 7)... See Table 16.1 from citation 29.

In addition, this genome has not been made publicly available yet, I checked GenBank and no genome assembly is available there.

*It is my foremost request that the authors MUST make all the data of their study, including the full genome and transcriptome data of the species investigated, as well as all MS raw files and code for analyses publicly available, for example in a GitHub repository, or on Data Dryad or Figshare. Without all underlying data being used made available according to the FAIR principles, the study cannot be repeated and the validity assessed.

*In addition, providing sequences and their alignments only as Figures is useless; please provide all of them in a usable format, such as Fasta or Nexus; a GitHub (or similar) repository would be most appropriate for this. For example, Supplementary Figs. 1, 4, 6 need to be provided as a useable alignment.

AR: Please, note that this long comment, which is reported to the Editor as a major concern, is an issue that can be easily addressed.

Unlike it is stated by reviewer 3 here and in other comments below, we have no problem with making ALL DATA available and do not understand why reviewer 3 seeks to introduce the opposed idea several times in his/her report. The genome of *Euplectella curvistellata* was made publicly available upon submission of our manuscript to Nature Communications, at the Tottori University Research Result Repository, Japan (Link: <https://repository.lib.tottori-u.ac.jp/14024>). In the two previous papers indicated by the reviewer, which dealt specifically with protein glassin,

only the sequences of interest, and not the whole genome, were released because other important studies (like this one) were in progress. With the submission of the current manuscript, we have released the complete genome. Our apologies for not being able to do so earlier, but we hope the reviewer will understand the situation.

In addition, the reviewer is unfairly criticizing us when stating that we are providing sequences only as figures. Please, check our supplementary materials, particularly the Source Data Files to corroborate that all alignments used in this study are provided in Fasta format.

The reviewer is also negatively prejudging the quality and significance of *E. curvistellata* genome. We already indicated in our Result section that the BUSCO value for this genome is a bit lower (64.1%) than the average of the remaining hexactinellids (68.7 %), but the genome is still largely informative, especially since only 7 genomes of hexactinellid sponges are available to date. It is not fair to state that “the genome has a questionable quality and that no wonder why glassin could not be retrieved from this “genome” in full”. Please, note that such a genome allowed in a first previous study to characterize the sequence of glassin (2015) and, in a second study (2018), to understand which domains are involved in silica polymerization, contributing with two giant steps to our current understanding of silica formation in Hexactinellida. It has also revealed eight different hexaxilins in *E. curvistellata*. So, we do not really see where the problem is.

2) Ehrlich and colleagues published on the presence of several other components of (hexactinellid) sponge spicules, i.e., chitin (polysaccharide), collagen, and most recently actin. These findings are completely ignored in the Introduction, and barely mentioned/discussed in the Discussion.

I would like to see a more comprehensive treatment of the issue. Especially the puzzling occurrence of actin in the glass sponge spicules studied by Ehrlich et al. (DOI: 10.1002/advs.202105059) needs consideration; actin had a molecular weight of 42 kDa, but I do not see a band here in Fig. 2 at that size – did it slip through the sample preparation?

A prominent target for investigation would be the axial filament, which Shimizu et al. claim to be Hexaxilin, and Ehrlich et al. to be actin. So, there is some discrepancy here that needs to be resolved, for example by co-staining with antibodies – Ehrlich et al. used a primary anti- β -actin antibody and secondary (anti-rabbit IgG (H+L), F(ab')₂ Fragment (Alexa Fluor 488 Conjugate) antibodies, so these analyses should be repeated with your species.

One problem here might be the differences in the sample preparation: in the present study the authors used HF treatment to dissolve the silica spicules, in some of Ehrlich et al.'s work for the detection of collagen and chitin they used NaOH, but for the detection of actin also HF.

So, I suggest, for better comparability, that the authors also prepare the organic fraction of the spicules with NaOH instead of HF and analyse the protein fraction remaining. I find it quite puzzling that Shimizu et al. did not detect any collagen, chitin, or especially actin (which was detected by Ehrlich et al. 2022 by also dissolving the spicules with HF). This really needs detailed addressing.

As it stands now, the reader is really left confused with what is going on with silicification in hexactinellid sponges; I think the authors need to bring all this together.

AR: We are mentioning actin and chitin in the Introduction of the revised version, then redirecting readers to supplementary text containing more detailed explanations (see new Supplementary Text-2). But, please, note also that our study is about the evolution of silicifying proteins in Hexactinellida. Chitin and actin are neither silicifying proteins nor, in the current stage of knowledge, can they help to decipher the evolution of silicification in Hexactinellida, being widely distributed in all sponge classes.

Anyway, in response to this comment by the reviewer we have included a new paragraph of Main Text dealing with actin, a new Supplementary Fig. 1 (see Response Fig. 2 in our above response to reviewer 2), and two new sections of Supplementary Text (-2 and -3). Essentially, we are indicating that traces of actin were not found in any our SDS-PAGE and Western blot analyses and discussing what the reasons could be.

Please, note that our SDS-PAGE and Western blot analyses of silica extracts, revealed well-marked bands of glassin, perisilin and hexaxilin, as corresponding to major protein components of the studied spicules (See Fig. 1a, c, e at the bottom of this letter, which will become new Supplementary Fig. 1). Additionally, we found some other —much fainter— protein bands in the gels, all of them with molecular weight higher than 70 kDa (see red arrows in Fig 1e at the bottom of this letter). We could not obtain sequence information from those minor bands, because their abundances in the gels are consistently low. Since actin is a 42 kDa protein, it can be ruled out that any of those minority protein bands of high molecular weight could be actin.

However, our negative finding regarding actin does not disprove that actin could actually be there. Why our approach aimed at revealing the silicifying proteins does not capture the presence of actin in hexactinellid silica, unlike that reported by Ehrlich et al.¹? As summarized in the new Supplementary Text-2, there are important differences between our methodological approach aimed at identifying silicifying proteins and that by Ehrlich et al. aimed exclusively to detect actin. First, we performed a more aggressive cleaning of the outer organic matter of the spicules by immersion in bleach and then in concentrated HNO₃/H₂SO₄ (1:4) at room temperature, whereas Ehrlich et al. immersed the spicules in only 70% HNO₃ at room temperature for 3 days. Based on our previous experience with sponge silica, cold nitric acid may not be sufficient to completely remove all external traces of organic components. On the other hand, our more aggressive cleaning may make it easier for bleach and/or acids to infiltrate between the loose silica layers characterizing many of the hexactinellid spicules² or entering through cracks in the silica that lead to the axial canal, posing in greater risk the integrity of some of the proteins of the axial filament. We also conducted a much more aggressive digestion of the silica by complete immersion of the cleaned spicules in HF/NH₄F (2M:8M for 2-3 days, whereas Ehrlich and co-workers¹ used only 10% HF (~5 M) droplets falling on the spicules for 7h to 10h. Another major difference in the approach is that we used the extracts from the total silica digestion, whereas Ehrlich and co-workers used the extracts from axial filaments previously released from the silica. The process of concentration of the proteins in the extracts was also markedly different. Ehrlich et al.¹ were only able to retrieve actin (Ehrlich, Pers. Comm.) when the soluble extracts of the axial filament “were precipitated with ice cold acetone, incubated at -20°C and centrifuged at 10,000 g for 10 seconds” to obtain protein pellets. The need for such precipitation steps suggests that actin was in low concentration in those filaments. We did not perform such a precipitation step. For protein concentration in the soluble fraction, we used ultrafiltration. Theoretically, ultrafiltration would also concentrate actin, but, in practice, actin could also have been lost from our soluble fraction by selectively binding to the ultrafiltration membrane or by precipitating during ultrafiltration. There are also differences in electrophoresis approaches. We used NOVEX NuPAGE system, in which proteins always run at neutral pH conditions. In contrast, proteins are subjected to alkaline pH during electrophoresis in the conventional SDS-PAGE used by Ehrlich et al.¹. The NuPAGE system gives better resolution than the conventional system and, in fact, glassin does not appear in the conventional SDS-PAGE system. In this regard, we do not know whether actin might also behave unusually in the NuPAGE system. In summary, due to the aforementioned methodological differences, the absence of actin in our gels cannot be used to decide whether this protein is actually absent or present in the silica of *E. curvistellata* and *V. pourtalesii*.

The reviewer should also understand that the primary objective of our study is to unravel the evolution of silicifying proteins. Ehrlich’s study (page 8 of 10) openly states that actin has no direct involvement in silicic acid polymerization. Rather, it is hypothesized that F-actin would serve as a template around which the units of the silicifying proteins (i.e., silicatein) would be assembled to build the silicifying axial filament. Please, note that your suggestion to research on actin, in order to be relevant, would have to involve not only hexactinellid sponges, but also members of Demospongiae and Homoscleromorpha. Of course, that would be a massive work needing a manuscript (and funding) of its own, and falling far beyond the scope of our current study, which is already quite complex.

3) While the text reads in large parts OK, there are sections where the writing is difficult to follow; one example is the abstract, which needs a thorough proofread, as does the whole manuscript.

AR: We have re-checked the entire manuscript for language but, again, this opinion on language use comes into contrast with the opinions expressed by the two other reviewers. We kindly request the Editor to assess himself whether the manuscript requires an additional proofread.

More detailed comments and suggestions:

ABSTRACT:

As mentioned above, needs complete overhaul in terms of language.

AR: We have revised the Abstract as suggested.

L20: Sponges are not the oldest animal lineage; they are as old as their sister group.

AR: OK. We have reworded as “an early-diverging lineage”.

L23: “confirm” is a very strong word, I do not see (esp. due to a lack of in-depth characterization and consideration of the Homoscleromorpha) such a confirmation. Should be softened to “suggest”.

AR: OK. We have changed “confirm” to “suggest”, but our findings are “demonstrating” rather than “suggesting” that the three classes of siliceous sponges (Hexactinellida, Demospongiae, and Homoscleromorpha) use independent protein machineries to build their skeletons. We are making clear in our responses below and in the revised manuscript that there is plenty of available (and unpublished) information supporting that Homoscleromorpha silicify using proteins other than silicatein, hexaxilin, glassin and perisilin. Please, see also Response Figure 1 above, which contains unpublished information on new silicifying proteins recently discovered (yet unpublished) by our group.

L31: There are no “higher” metazoans. This is incorrect phylogenetic terminology that puts vertebrates / humans at the top – we are past those days. Why not be more specific and replace with “bilaterians”?

AR: Lower and higher metazoans are common terminology, but we accept the comment and have changed the term to “bilaterians”, as it is suggested by the reviewer.

INTRODUCTION:

Some language improvements needed.

P2 L2: consider adding “extant” to Porifera, because, according to Systema Porifera, there is another extinct class of the Porifera, the Archaeocythida.

AR: We have now made it clear that we meant “extant” Porifera. Thank you.

P3 L11/12: unclear to me what you mean with “dissolved condition”. Consider to explain this better.

AR: Ok. Reworded: “The solvable nature of glassin also made it difficult to explain how such a protein could constitute the solid rod (i.e. the axial filament) that is observed within hexactinellid spicules and that is assumed to determine spicule shape during silicification”.

P3 L14: Here you should provide more information about other recent findings of organic components isolated from Hexactinellid spicules, such as chitin, collagen, and actin – see Herman Ehrlich’s work (and also see comment above).

AR: The Introduction of our research subjects (silicifying proteins) is recommended to be developed in about 500 words. Chitin, collagen and actin are not silica-polymerizing proteins and for that reason, we cannot extend more on them in the Introduction. However, in response to this comment, we have prepared a new section of Supplementary Text-2. We hope the reviewer finds it useful.

P3 L22: Change “the findings provide novel mechanistic insights” to “the findings suggest novel mechanistic insights”. After all, without functional experiments you cannot really prove function (the mere presence of a protein does not necessarily prove its function, it just suggests involvement).

AR: We have reluctantly removed the word “mechanistic”. However, please, note that the findings allow us to discuss on some aspects of the mechanism of silicification. Hexaxilin appears to act intracellularly whereas perisilin and glassin appear to act extracellularly. This division of labor between different and unrelated proteins represents a major advanced in knowledge. Until now, the only hypothesis was that different isoforms of just silicatein operated throughout the silicification process. In that sense, our findings introduce a new perspective on both the evolution of the silicification but also the mechanics of the process.

P3 L24: replace “higher metazoans” (see comment above) with “bilaterians”

AR: OK.

RESULTS & DISCUSSION

P3 L34: Where is the genome accessible at? Without the genome publicly accessible and assessable the study cannot be replicated and the findings validated. As mentioned above, making the genome public is a non-negotiable requirement for publication of this study.

AR: The transcriptome of *Vazella* was made publicly available in 2019 upon its publication in *Science Advances* and the genome of *Euplectella* was made available upon the initial submission of this manuscript to *Nature Communications* at the link <https://repository.lib.tottori-u.ac.jp/14024>.

P3 L29 – P4 L20: I am puzzled that the authors did not detect at all in their spicules any other biological components that were suggested to be present in the Hexactinellida, such as chitin, collagen, and action (see comments above). If so (no traces) this should explicitly be stated here somewhere in this section.

AR: We have already addressed this issue extensively in a previous response to the reviewers and in new Supplementary Text-3 and Supplementary Fig. 1.

P4 – P10: The subsections on Hexaxilin, Persilin, and Glassin are very long, wordy, and in parts difficult to read/follow. Consider shortening and/transferring parts into the Supplementary Material.

AR: We have revised and re-organized the text of those sections to our best to improve readability, but not shortened sections, as reviewers 2 suggested additional analyses that had to be incorporated necessarily.

P5 L15: “is acceptable for Porifera”. Unclear and not well substantiated statement. What is this based on, how is this judged, where is the threshold? Do you have a reference for this? Otherwise, rephrase please.

AR: OK. We have rephrased using comparative terms: “Occurrence of HXX-2 to HXX-8 only in Hexasterophora and their absence in Amphidiscophora may reflect a phylogenetic pattern, but it may also be an artifact, since the quality of transcriptomes and genomes of the six Hexasterophora species is comparatively better (BUSCO completeness: 64%-81%) than that of the transcriptome (36%) of the Amphidiscophora species”.

P6 L5: I am not really convinced by your reasoning about the origin of the MBLf gene through horizontal gene transfer. I think a more in-depth analysis of this gene in both the Hexactinellida and the bacteria is in order for a better substantiation of this claim. Have other cases of horizontal gene transfer been reported and substantiated in the Porifera?

AR: We agree now with the reviewer, since after checking EukProt database, five hexaxilin-like hits have been retrieved within Protista. That information was not available in the various other databases previously examined, which falsely suggested a plausible lateral gene transfer. The four

protist hits are all marine organisms. One is an Apicomplexa, the gregarine *Siedleckia nematoides*, which parasitizes polychaetes. The four remaining hits concentrate within the supergroup Rhizaria (Cercozoa + Foraminifera + Radiolaria), that is, protists emitting pseudopodia. Three hits occur in Cercozoa (the chlorarachniophyte algae *Bigelowiella natans* and *Amorphocloro amoebiformis*, and the giant amoeba *Gromia sphaerica*). The fourth rhizarian is the foraminifer *Bolivina argentea*. Surprisingly, no other protist groups contain a hexaxilin-like sequence homolog to that hypothetically inherited by metazoans from their protist ancestor. Also surprisingly, Radiolaria — the only silicifying rhizarians — lack hexaxilin-like sequences. We have summarized all this information in the revised version and conducted a new phylogenetic analysis incorporating the new hits in order to produce the most updated tree for the revised manuscript (see new Fig. 4 and Supplementary Fig. 3). Although the evolutionary significance of these silicifying protein does not change at all for Hexactinellida, hexaxilin cannot be further considered the result of lateral gene transfer, but rather as resulting from a duplication of a sponge hexaxilin-like gene inherited from its protist ancestors.

P7 L7 “Because homosclerophorid sponges lack silicatein and hexaxilin, ...” Do they? Did you analyse any homosclerophorid sponges with the same methods? Or do you have a reference for this statement where this has been done?

AR: At the present stage of knowledge, doubts about the absence of silicatein, hexaxilin, glassin and perisilin in Homoscleromorpha are not justified. The absence of silicatein in Homoscleromorpha was already demonstrated by the transcriptome of *Corticium candelabrum* (Riesgo et al. 2015) and more recently by the genome of *Oscarella lobularis* (publically available at Sanger). For this manuscript, we re-analyzed Riesgo’s transcriptome from scratch. For the revised manuscript, we have also examined the newly released, high-quality genome of *O. lobularis*. Both homosclerophorid species lack silicatein (but not capthesin L-like enzymes), hexaxilin, perisilin, and glassin. Therefore, there is no doubt: those silicifying proteins do not occur in Homoscleromorpha. Consequently, all three sponge classes use independent protein systems for silicification, as we are reporting.

Furthermore, we have even additional (unpublished) evidence that Homoscleromorpha sponges use for silicification protein systems different from those in demosponges and hexactinellids. We can state it with absolute certainty because we have been investigating silicification in Homoscleromorpha during the past 6 years. At the moment, we have identified a new protein extracted from the peripheral silica (preliminarily named by us as sclerogen; see unpublished information in Response Figure 1 in this letter). These days, we are attempting to characterize a second new protein (still nameless). This latter protein occurs at the axial filament and is different from the sclerogen extracted from the peripheral silica (and the axial protein is not actin either). Thus, in addition to the already published information indicating the absence of silicatein in Homoscleromorpha, our most recent, unpublished findings further corroborate that silicification in homosclerophorid sponges is mediated by proteins other than silicatein. As you can see in Response Figure 1, we have developed specific antibodies against sclerogen and conjugated them with nanogold. Note in the picture to the left that nanogold is labelling traces of sclerogen protein in the peripheral silica of the spicule but not at the central axial filament (AF), which is made of a different —still uncharacterized— protein, which is also different from actin. None of those two homosclerophorid proteins are related to other silicifying proteins known from Porifera to date. We are herein disclosing this unpublished information to show the Editor and reviewer 3 that there is NO DOUBT that Homoscleromorpha sponges lack silicatein. Thus, we must insist, our main message is very solid: **all three sponge classes use independent protein systems for silica polymerization.**

P7 L16: “of the ancestral ocean where the major sponge classes arose” (citation 15). Please provide more details here about when this happened, and what the likely ocean conditions were

at that time. I suggest to cite a recent molecular clock study here that specifically deals with the timing of the divergences of the animal lineages.

AR: As suggested, we have replaced this brief mention with more detailed explanations that have resulted in a new section of the main text (Origin and evolution of silicification) with important derivations that reconcile the current mismatch in dating between molecular clocks and the fossil record.

P8 L9-11: This sentence is not well phrased and is largely speculation. It needs more evidence.

AR: Rephrased as: “It is worth noting that none of the seven hexactinellid taxa studied has all ten perisilins (Figs. 3, 7). Perisilin-6 occurs in five species from five different families, but most other perisilin types occur in only one, two or three related species. This distribution pattern suggests that perisilins would be involved in producing skeletal features restricted to groups of related species (e.g., genus, family), making those groups skeletally distinguishable from others. However, it is not yet clear whether all identified hexactinellid perisilins can silicify.”

P9 L7: again, make the genome publicly available; you refer here to a specific contig in the genome but that genome is not available.

AR: As explained in previous answers, it is already publicly available.

P9 L30 – 36: this whole section reads quite confusing, please rephrase to make clearer. How does that impact your narrative if Glassin is absent from *V. pourtalesii*?

AR: We have re-checked the section, but we do not really understand this question. How could we discuss about assumptions that are not real? These are the facts: Unlike for hexaxilin and perisilin, a single glassin sequence occurs in each of the hexactinellid species investigated, also in *Vazella*. However, *Vazella* glassin is not upregulated in high silicate availability. Also, glassin does not show up in the extracts of *Vazella*'s silica. Neither is the silica of *Vazella* labelled by a specific antibody developed by us from the sequence of the *Vazella*'s glassin. These three pieces of experimental evidence (which is not little) strongly suggests that glassin is not silicifying in *Vazella* (as it appears to be also the case in *A. vastus*, *R. fibulata*, and *S. nux*). Besides, we are literally recognizing in our text that “the overall picture is still blurry”. Thus, we consider that there is nothing else of this topic that can be further discussed without being redundant.

P10 First paragraph of that section. The value of this section (and the associated TEM Figure 9) is elusive to me. How does that add to the interpretation of the data? In Fig. 9 no axial filament (or other proteinaceous components) is visible in the TEMs. To make such an ultrastructural study really useful to strengthen the case, you would need to carry out immunogold labelling to identify the location of the expression of, e.g., hexacilin. This would be of great value. Otherwise omit these TEMs, or shift them to the supplementary material.

AR: As you can see in our unpublished Response Fig. 1, we are familiar with the use of nanogold for locating silicifying proteins in Porifera. However, a number of reasons render such an approach unfeasible for *Vazella* and *Euplectella* (and most other hexactinellids): 1) The chamber where the high pressure freezing of the tissue is conducted host only 200 µm-sized samples; it means that most hexactinellid spicules would not even fit; 2) More importantly, to obtain workable hexactinellid tissue on which conduct the ultramicrotomy and the subsequent antibody incubation for electron microscopy, the large hexactinellid spicules need to be eliminated from the tissue; it means that the silica will disappear along with all the associated proteins in both the peripheral silica and the axial filament. We know this well because we have already attempted it and it is also the reason why you cannot see axial filaments in our figure (which was already explained in the original manuscript). 3) Another important reason why nanogold labelling cannot be easily conducted with *Vazella* and *Euplectella* is because these two species are bathyal sponges growing in off shore habitats, which are logistically very hard to reach. For our experiments on differential gene expression with *Vazella*, we had the support of a 10 million euros H2020 grant (SponGES), which allowed an international consortium to flight a working-class ROV (ROPOS) from the West to the East Canadian coast and to rent a big oceanographic vessel for 10 days of off-shore navigation and deep-sea work, which also included sophisticated equipment to keep the

sponges alive on board and in land laboratories for further experimentation, etc. As a result of all that work, we obtained much of the molecular information, presented in part herein. Now that we know all this new information, it would be great to repeat the journey and attempt immunolabelling experiments (as suggested by the reviewer)... and much more. But this is unlikely to be feasible in the next few years, unless we get another multinational, interdisciplinary, millionaire grant. Thus, in stating that “*you would need to carry out immunogold labelling to identify the location of the expression of, e.g., hexacilin*”, we believe that the reviewer is not fully aware of the economic and scientific magnitude of what is being requested.

In addition, the reviewer is questioning the value of figure 9 because axial filaments are not seen. To our knowledge, this is only the second time that scleroblasts of adult hexactinellids are described by TEM. Thus, the images may have some value, among other things because they testify about both similarities and differences regarding the silicifying cells of demosponges and homoscleromorphs: the occurrence of silicification vesicles, the presence of abundant electron-clear vesicles (filled with silicate?), the syncytial condition of scleroblasts, the plugs for connection to other cell types, the large syncytium size, etc. We did not get any criticism about this figure by the other reviewers, who appeared to understand the importance of the images, which—we must insist—are rarely available in the literature. Thus, being aware of the value of this figure for the future work of others, we prefer to leave it where it is, unless the Editor indicates otherwise.

P10 L19 (second paragraph): it would be really useful to provide a summary figure with a sketch of how you envisage the silicification process in the Hexactinellida, and what is different to the Demospongiae.

AR: Initially we also thought of some summary figure, but it cannot be accomplished successfully. The reason is that many details about how scleroblasts interacts with proteins and spicules are still unknown. Are the scleroblasts that form the axial filament a different cell lineage from those that carry out peripheral silicification for spicule thickening? Do the latter have a silicification vesicle that they would not use, or do they, etc, etc? There are still too many unknowns along the silicification process of hexactinellids, which will force us to draw what we do not know and cannot even deduce.

P10 last paragraph of that page and the first paragraph of P11: you treat the issue of the other “non-silicifying” proteins (like collagen and actin) very superficially, without much depth. You will need to provide evidence (see also comment above) that the two proteins are not present in the species you studied, and if you indeed detect them, how they interact with the proteins you discovered. Either by antibody labelling, as suggested above, or by in situ hybridization. Without this, your study is incomplete.

AR: The subject of actin has already been extensively addressed in our previous responses.

P11 L12: the diatoms appear quite sudden and random. Needs better integration.

AR: We mentioned diatoms briefly, just as an enriching, comparative example because they are, in quantitative terms, the most important silica producers. However, we do not see how we could best integrate them in our Discussion better, since the issue is a very co-lateral to our main subject and the word counts highly limiting. Please, note that we could also discuss in more detail on what is known from the silicification systems of other silicifiers, such as radiolarians, chrysophyceas, silicamoebas, choanoflagellates, plants, and more, but our database searches have revealed that (to date) the silicifying proteins of sponges cannot be related to the silicifying systems of other well-known silicifiers, including diatoms. What would be the point of discussing only diatoms any further? In fact, it would require another entire manuscript of its own to compare the known and unknown of different groups of silicifying organisms, a review that would go far beyond the scope of this research on Hexactinellida. We also believe that extending non-essential side topics would add nothing relevant to the study, while substantially diverting readers’ attention from the main message.

CONCLUSIONS

Need complete overhaul, some of the statements there are too strong and insufficiently substantiated.

AR: We would have preferred a more precise indication about which are the specific problems, according to the reviewer criterion. We have revised carefully our conclusions and we consider that all statements are sufficiently substantiated, except for one point: the origin of hexaxilins and perisilins, which, after the EukProt hits, need to be reworded. Moreover, we have already indicated in the revised manuscript and in our responses that there is overwhelming, compelling evidence that all 3 classes of siliceous sponges have evolved their silicifying protein machinery independently. Thus, we cannot see where the problem is.

Figure4: Please provide species names in the tree at the leaves and no abbreviations (this applies to ALL trees!)

AR: OK. We have re-designed all trees, with improved labeling and extended species names to facilitate sequence identification.

The class is Demospongiae not Demospongia (correct in all trees) Generally: please provide more information about the hexaxilin discovered in the Calcarea; at present the explanation in the text is insufficient. Provide a deeper analysis of the genes discovered, what are the exact differences to the gene in Hexactinellida and Demospongiae, where they apparently are involved in silicification. Same applies to Persilin.

AR: Ok. Misspelling corrected. Thank you.

Our supplementary tables already contain many comparative data (including global identities, bit score values, total sequence length, etc). Nevertheless, in response to this comment and regarding hexaxilin-like proteins, we have prepared a new paragraph of Supplementary Text-5 (Hexaxilin-like sequences) and a new Supplementary Fig. 3 (Alignments of all hexaxilin-like sequences of Porifera; see Response Fig. 8 pasted below). In essence, we explain that “Hexaxilin-like sequences were identified in sponges from the classes Hexactinellida, Calcarea and Demospongiae (Supplementary Fig.3), but, in all cases, with very low identities (25-29%) and E-values larger than $6.0e^{-31}$, which casted doubts about function preservation and, in some cases, even about true sequence homology. Interestingly, the hexaxilin-like sequences of Calcarea make an independent, cohesive group that does not relate to hexaxilin-like sequences of either Hexactinellida or Demospongiae (Fig. 4). In contrast, hexaxilin-like sequences of Demospongiae make two independent groups, one related to those of Hexactinellida and the other related to those of Apicomplexa protists and non-sponge metazoans. In the legend of new Supplementary Fig. 3, we also indicate that “All sequences of Calcarea and Hexactinellida have six cysteine positions conserved, whereas, among Demospongiae, variability occurs in *Kirpatrickia variolosa*, *Latrunculia apicalis*, *Amphimedon queenslandica* and *Ephydatia muelleri*. Three of the seven putative Zn^{2+} - binding sites are conserved (positions 278, 279 and 536) and the four others show amino acids different from those in hexaxilin-1 of *E. curvistellata*. Note that several hexaxilin-like sequences of Calcarea, Demospongiae and Hexactinellida show long amino acid insertions compared to hexaxilin-1 of *E. curvistellata*, which may change protein conformation and/or function.”

Regarding, perisilin-like sequences, we have also prepared a new Supplementary Fig. 8 (see Response Fig. 9 below), comparing the alignment of perisilin-like proteins in Porifera to the sequence of perisilin-1 of *Vazella pourtalesii* (Vp_1621.i8). In the caption, we indicate that “all perisilin-like sequences show nine cysteine conserved positions out of the twelve cysteine conserved positions occurring in hexactinellid perisilins. No histidine-rich region related to silicification is present in any perisilin-like sequence of Calcarea, Demospongiae or Hexactinellida.” There is nothing remarkable in terms of sequence insertions/deletions. Nothing much can be deduced about the relationships of the poriferan perisilin-like proteins because they are distributed in small clusters with unreliable node support, as does most of the phylogenetic structure of the outgroup of perisilins: a large a polytomy (see Fig. 7 in the revised manuscript).

We hope these new additions are satisfactory to the reviewer.

Response Figure 8 (=Supplementary Fig. 3). Alignments of hexaxilin-like sequences of Porifera. Conserved cysteine positions and predicted Zn²⁺ binding positions in hexaxilin-like proteins of Calcarea (blue), Demospongiae (purple) and Hexactinellida (orange) are compared to Hexaxilin-1 of *Euplectella curvstellata* (Ec_20423). The sequences were aligned using MAFFT-online. Sequence codes as in Fig. 4, Supplementary Fig. 4 and Supplementary Table S1. All sequences of Calcarea and Hexactinellida have six cysteine positions conserved, whereas, among Demospongiae, variability occurs in *Kirpatrickia variolosa*, *Latrunculia apicalis*, *Amphimedon queenslandica* and *Ephydatia muelleri*. Three of the seven putative Zn²⁺ binding sites are conserved (positions 278, 279 and 536) and the four others show four amino acids different from those in hexaxilin-1 of *E. curvstellata*. Note that several hexaxilin-like sequences of Calcarea, Demospongiae and Hexactinellida show long amino acid insertions compared to hexaxilin-1 of *E. curvstellata*, which may change protein conformation and/or function.

Response Figure 9 (=Supplementary Fig. 8). Alignments of perisilin-like sequences of Porifera. Conserved cysteine and histidine positions in perisilin-like proteins from Calcarea (blue), Demospongiae (purple) and Hexactinellida (orange) are compared to perisilin-1 of *Vazella pourtalesii* (Vp_1621.i8). The sequences were aligned using MAFFT-online. Sequence codes as in Fig. 7, Supplementary Fig. 9 and Supplementary Table S5. All perisilin-like sequences show nine cysteine conserved positions out of the twelve cysteine conserved positions occurring in hexactinellid perisilins. No histidine-rich region related to silicification is present in any perisilin-like sequence of Calcarea, Demospongiae or Hexactinellida.

REFERENCES USED IN EXPLANATIONS TO THE REVIEWERS

1. Ehrlich, H. *et al.* Arrested in glass: Actin within sophisticated architectures of biosilica in sponges. *Adv. Sci.* 2105059 (2022) doi:10.1002/advs.202105059.
2. Maldonado, M. *et al.* On the dissolution of sponge silica: Assessing variability and biogeochemical implications. *Front. Mar. Sci.* **9**, 2487 (2022).
3. Maldonado, M. *et al.* On the dissolution of sponge silica: Assessing variability and biogeochemical implications. *Front. Mar. Sci.* **in press**, (2022).
4. Riesgo, A., Maldonado, M., López-Legentil, S. & Giribet, G. A proposal for the evolution of cathepsin and silicatein in sponges. *J. Mol. Evol.* **80**, 278–291 (2015).
5. Shimizu, K. & Morse, D. E. Silicatein: A unique silica-synthesizing catalytic triad hydrolase from marine sponge skeletons and its multiple applications. in *Methods in Enzymology* vol. 605 429–455 (Academic Press Inc., 2018).
6. Shimizu, K. *et al.* Exploration of genes associated with sponge silicon biomineralization in the whole genome sequence of the hexactinellid *Euplectella curvistellata*. in *Biomining* (eds. Endo, K., Kogure, T. & Nagasawa, H.) 147–153 (Springer, 2018). doi:10.1007/978-981-13-1002-7_16.
7. Maldonado, M. *et al.* Cooperation between passive and active silicon transporters clarifies the ecophysiology and evolution of biosilicification in sponges. *Sci. Adv.* **6**, eaba9322 (2020).
8. Francis, W. R. *et al.* The genome of the reef-building glass sponge *Aphrocallistes vastus* provides insights into silica biomineralization. *R. Soc. Open Sci.* **10**, (2023).
9. Santini, S. *et al.* The compact genome of the sponge *Oopsacas minuta* (Hexactinellida) is lacking key metazoan core genes. *BMC Biol.* 2023 211 **21**, 1–21 (2023).

10. Antcliffe, J. B., Callow, R. H. T. & Brasier, M. D. Giving the early fossil record of sponges a squeeze. *Biol. Rev.* **89**, 972–1004 (2014).
11. Maliva, R. G., Knoll, A. H. & Siever, R. Secular change in chert distribution: a reflection of evolving biological participation in the silica cycle. *Palaios* **4**, 519–532 (1989).
12. Maliva, R. G., Knoll, A. H. & Simonson, B. M. Secular change in the Precambrian silica cycle: Insights from chert petrology. *Geol. Soc. Am. Bull.* **117**, 835–845 (2005).
13. Siever, R. Silica in the oceans: Biological-geochemical interplay. in *Scientists on Gaia* (eds. Schneider, S. H. & Boston, P. J.) 287–295 (MIT Press, 1991).
14. Siever, R. The silica cycle in the Precambrian. *Geochim. Cosmochim. Acta* **56**, 3265–3272 (1992).
15. Nelson, D. M., Tréguer, P., Brzezinski, M. A., Leynaert, A. & Quéguiner, B. Production and dissolution of biogenic silica in the ocean: revised global estimates, comparison with regional data and relationship to biogenic sedimentation. *Global Biogeochem. Cycles* **9**, 359–372 (1995).
16. Garneau, A. P. *et al.* Aquaporins mediate silicon transport in humans. *PLoS One* **10**, e0136149 (2015).
17. Marron, A. O. *et al.* The evolution of silicon transport in eukaryotes. *Mol. Biol. Evol.* **33**, 3226–3248 (2016).

REVIEWER COMMENTS

Reviewer #2 (Remarks to the Author):

I've read through the updated manuscript and the author's response to reviews, and I think it is a very exciting manuscript that makes an important contribution to the field. There is no doubt that this manuscript should be published. With that said, I am not entirely satisfied with all of the author's responses. These are not issues that preclude publication, but I will restate my objections for the authors to consider.

A minor point - there is still some language in the manuscript like 'higher metazoan' that should be removed. Also, the term Protista is used in several places, and this is not a formal taxonomic name as it represents a polyphyletic group. I encourage the authors to use 'protists' or 'microeukaryotes' in place of this term.

I'm a little confused about the authors use of the term 'isoform' when referring to α , β , and γ persalin. Isoforms refers to proteins that are encoded by the same gene but differ due to alternative splicing of common transcripts prior to translation, or due to post-translational processing (e.g., cleavage). Thus, isoform would indicate that these are encoded by the same gene. If they are encoded by related genes at different position in the genome, they would be paralogs. The nature of these 'isoforms' is unclear to me from the text, and without a very complete genome it might be hard to determine the source of protein variants as being from splice variation, allelic variation, or post-translational processing?

My prior comments about the appropriateness of the proteomic methods have not been adequately addressed (and seem to have been misunderstood by the authors). I was less concerned about the authors' ability to test the claim that actin is involved in spicule formation, and more concerned that the use of edman degradation on protein bands isolated after gel separation needlessly reduced the number and diversity of spicule-derived proteins that were detected. I understand the authors' point that this methodology has been used successfully in the past, and indeed it was used successfully here to identify novel spicule associated proteins. Still, as the authors note, there are additional bands visible in their gels which they state are too low in abundance to sequence by Edman degradation. This is where more modern LC/MS-based proteomic methods are superior. If they can see bands on a gel, they could absolutely analyze the entire sample (without even needing to cut bands out of a gel or membrane), and with reference to the transcriptome/predicted proteome should easily be able to determine the identity of those bands. This is neither difficult, expensive, nor time consuming and would strengthen the study.

Likewise, I was not convinced by the author's description of their antibody validation methods (which, still seem to be excluded from the manuscript and are only in the response to reviews - albeit the supplemental materials are extensive so I may have missed it). 1) The authors have now included western blot results to demonstrate the specificity of their antibodies. In the case of Euplectella hexaxilin, the result is reasonably convincing. With respect to glassin, much less so - there is a huge amount of protein on loaded on the gel, and very faint staining which suggests to me the antibody does not really bind with high affinity to glassin, at least denatured glassin which is surprising considering that it is a peptide antibody. Also, here and in other blots, the authors consider higher molecular weight bands to be oligomers, but this is speculation and it is unclear why they would have oligomers if reducing agents were added to disrupt disulphide bonds. 2) the Western Blots for Vazella hexaxilin and persilin are very unconvincing. Indeed, there are multiple bands

detected, and the authors tell us what they think those bands could be, but there is no way of knowing with any certainty. 3) The only other validation measure presented are Elisa results from Sigma after antibody production. These only show that the rabbits did produce antibodies against the injected immunogens but not that the antibodies are specific and recognize the endogenous, folded protein in the sponges.

It remains unclear why the authors wouldn't affinity purify their antibodies against the injected antigen, and despite clear and interpretable immunostaining patterns there remains uncertainty about whether these antibodies are specific to the intended endogenous target.

Regarding immunostaining patterns and faint peripheral staining of hexaxilin, the authors suggest that this is due to antibody getting trapped in spicule fractures and being inadequately washed out. This is possible, but there is no evidence to support this interpretation. In fact, if this interpretation were correct, then the pre-immune sera control and/or secondary-only control should give the same pattern, but this is not shown. I think it is at least possible that hexaxilin is present at lower abundance in peripheral areas of the spicule too.

I stand by my opinion that this study is exciting and important. I've highlighted the weaknesses that I believe remain, but these should not prelude publication of the manuscript.

RESPONSES TO REVIEWER 2

I've read through the updated manuscript and the author's response to reviews, and I think it is a very exciting manuscript that makes an important contribution to the field. There is no doubt that this manuscript should be published. With that said, I am not entirely satisfied with all of the author's responses. These are not issues that preclude publication, but I will restate my objections for the authors to consider.

AUTHOR RESPONSE: We thank the reviewer for his/her kind words and recognition of the significance of our study. We would also like to thank this reviewer for providing constructive, valuable comments that undoubtedly helped us to make a stronger manuscript. In this new revised version (R2) of the manuscript and in our response letter, we have again made our best to address and strengthen the possible weaknesses indicated by the reviewer. Yet, we must also insist that there are no specificity problems with our antibodies. We are explaining why through this response letter, using both information published in the literature and our own data, a body of evidence that supports our approach.

A minor point - there is still some language in the manuscript like 'higher metazoan' that should be removed. Also, the term Protista is used in several places, and this is not a formal taxonomic name as it represents a polyphyletic group. I encourage the authors to use 'protists' or 'microeukaryotes' in place of this term.

AUTHOR RESPONSE: Sorry that we forgot to replace some of those terms with their most suitable version. It is done now.

I'm a little confused about the authors use of the term 'isoform' when referring to α , β , and γ peralins. Isoforms refers to proteins that are encoded by the same gene but differ due to alternative splicing of common transcripts prior to translation, or due to post-translational processing (e.g., cleavage). Thus, isoform would indicate that these are encoded by the same gene. If they are encoded by related genes at different position in the genome, they would be paralogs. The nature of these 'isoforms' is unclear to me from the text, and without a very complete genome it might be hard to determine the source of protein variants as being from splice variation, allelic variation, or post-translational processing?

AUTHOR RESPONSE: We are afraid that the reviewer's concept of isoform is not up to date. There are many studies reporting proteins encoded by closely related but different genes (e.g., genes of the same family) to be isoforms. This is the well-known example of actin isoforms in mammals, for which studies report textually that "*Although actin is often thought of as a single protein, in mammals it actually consists of six different isoforms encoded by separate genes*" (Perrin and Ervasti, 2010; Gunning and Hardeman, 2018). We have reworded our text regarding this issue to make it more clear: "Comparatively, demosponges species have a smaller number of spicule types, and the different types are not produced by the participation of proteins from several unrelated genes, but only by silicatein isoforms, which have different expression patterns (Müller et al., 2007). For instance, in the fresh-water demosponge *Ephydatia fluviatilis* the various isoforms are produced by expression of up to ten closely related *silicatein* genes (Mohri et al., 2008)". Note that in that sentence we are reporting what has been published in the literature and is widely accepted; it is not our personal interpretation or our own data.

Through our experimental results, the term "isoform" has been used to refer to those proteins that, according to the transcriptome assembly of *Vazella pourtalesii* performed by Trinity's algorithms, are derived from very similar transcripts. They therefore putatively represent either the

result of alternative splicing or allelic variants of the same gene, as inferred by Trinity software. We agree with the reviewer that without a complete reference genome no one can be completely sure whether “very similar transcripts” derive from the same gene or from different but closely related genes. Such a genome is not available for *V. pourtalesii*. Therefore, we have to rely on Trinity algorithms for not only assembly but also for inference of putative isoforms. In short, we followed Trinity’s advice – What else could be done? What would this reviewer do in our situation? We had already explained our way of proceeding regarding isoforms in the Methods: ““Protein isoforms were identified and distinguished from related protein sequences whenever possible, following the isoform prediction criterion implemented in Trinity””.

My prior comments about the appropriateness of the proteomic methods have not been adequately addressed (and seem to have been misunderstood by the authors). I was less concerned about the authors’ ability to test the claim that actin is involved in spicule formation, and more concerned that the use of edman degradation on protein bands isolated after gel separation needlessly reduced the number and diversity of spicule-derived proteins that were detected. I understand the authors’ point that this methodology has been used successfully in the past, and indeed it was used successfully here to identify novel spicule associated proteins. Still, as the authors note, there are additional bands visible in their gels which they state are too low in abundance to sequence by Edman degradation. This is where more modern LC/MS-based proteomic methods are superior. If they can see bands on a gel, they could absolutely analyze the entire sample (without even needing to cut bands out of a gel or membrane), and with reference to the transcriptome/predicted proteome should easily be able to determine the identity of those bands. This is neither difficult, expensive, nor time consuming and would strengthen the study.

AUTHOR RESPONSE: We appreciate the proposal for adding new proteomic techniques. We will incorporate such an LC/MS approach into our ongoing research on *Homoscleromorpha silica* proteins for sure. However, we understand that it is too late for this manuscript to repeat and replace all our Edman-degradation proteomics by the more modern LC/MS approach. It would help to resolve minority protein bands that we could not resolve in this study (you cannot resolve everything in a study!). Nevertheless, it would also involve additional months of work, the addition of new co-authors, and complete rewrite of the manuscript if the proteins in the minority bands were finally characterized. At this point, we believe that the two new proteins discovered here, together with the reinterpretation of the role of glassin, already represent important conceptual advances both in terms of spicule silicification and the evolution of the Porifera skeleton. Even if some minority protein bands were solved, they would not make the conceptual message of the study more significant, because the main conclusion would remain unchanged: the three different classes of sponges have evolved independent protein machineries to produce their silica skeletons.

Therefore, in response to this point, we have added a comment into the Main Text indicating that the unresolved minority protein bands could be resolved in future studies using LC/MS-based proteomic methods: “Protein abundance in these bands was systematically low and did not allow any further protein identification through our Edman degradation approach. Future approaches using LC/MS-based proteomic methods might be more successful.”

We ask this reviewer to understand that not everything can be resolved in a single study. We will certainly use the LC/MS approach in future studies. In fact, we are currently gathering representatives of Amphidiscophora, also Lychniscosida and Sceptrulophora in Hexasterophora, with the objective of better understanding the diversity and distribution of proteins in the silica through the class Hexactinellida.

Likewise, I was not convinced by the author's description of their antibody validation methods (which, still seem to be excluded from the manuscript and are only in the response to reviews - albeit the supplemental materials are extensive so I may have missed it). 1) The authors have now included western blot results to demonstrate the specificity of their antibodies. In the case of *Euplectella* hexaxilin, the result is reasonably convincing. With respect to glassin, much less so - there is a huge amount of protein on loaded on the gel, and very faint staining which suggests to me the antibody does not really bind with high affinity to glassin, at least denatured glassin which is surprising considering that it is a peptide antibody. Also, here and in other blots, the authors consider higher molecular weight bands to be oligomers, but this is speculation and it is unclear why they would have oligomers if reducing agents were added to disrupt disulphide bonds.

AUTHOR RESPONSE: In addition to Supplementary Figure 1, elaborated in response to previous comments by the reviewer, we add here a new supplementary figure for anti-hexaxilin controls using pre-immune sera (see response to the last comment). We consider the additional antibody characterization methods by Sigma-Aldrich Japan (and in Japanese) to be standard company protocols that might be of little interest to readers if they were included as supplementary materials. This is why we only show them in the letter to the reviewers.

We are glad that the reviewer finds the evidence presented for hexaxilin convincing. We hope that, after this letter, the reviewer will also be more convinced that the other antibodies are also fine. As for glassin, the reviewer indicates that "there is a huge amount of protein loaded on the gel, and very faint staining which suggests to me the antibody does not really bind with high affinity to glassin". Please, note that such a pattern also has explanations other than antibody affinity, as we develop below. First, we must admit that our western blot gels are not the best ever. The reason is that we never thought that these western blots would ever be published. Rather, we performed them as internal, initial tests of our antibodies. In fact, the anti-glassin test was performed in 2014, while the anti-hexaxilin in *Euplectella* was run in 2017, and the anti-hexaxilin and anti-perisilin of *Vazella* in 2022, as successive findings added complexity to this study. We rescued some of these gels from our archives to prepare a new figure S1 for a first revised version of the manuscript, trying to fulfill the reviewer's requests in a timely manner. Over the years, not all of our gels were run under identical conditions as to provide NORMALIZED comparison. Therefore, they are not ideal for comparing staining intensity, nor making the semiquantitative interpretations intended by the reviewer. The reviewer is probably also aware that staining intensity can be further affected by factors others than antigen specificity, such as antibody concentration, reaction time, reaction temperature, insufficient (or excessive) blocking reagent and others. Furthermore, in the case of broad bands, it does not necessarily mean that the antibody recognizes non-specific bands, as some proteins migrate anomalously on SDS-PAGE as a result of being post-translationally modified or alternately spliced, leading to forms with slightly different molecular masses but containing the same epitope recognized by the antibody. See also Gilda et al. (2015) for different commercial antibodies against a same protein having a disparity of reactivity on the western blot gels. Thus, the pattern in the western blot of glassin and the comparatively poor antibody labelling may also be reflecting the wide dispersion of the protein in the gel, probably in combination with some overblocking. In addition, we suspect that glassin may bind to PVDF membranes inefficiently due to its low content in hydrophobic amino acids. Furthermore, different antibodies are known to exhibit different linear dynamic ranges and may not produce equally intense bands even if used at the same concentrations. There are examples of fold changes in protein expression detected to decrease with increasing protein loading (Charette et al., 2010). Despite all these possibilities and considerations regarding non-normalized comparisons, obvious bands still appear on all our gels, indicating that the antibody for glassin and those for the other proteins are specifically recognizing the target proteins. Note also that those antibodies were not created using whole proteins as immunogens

but rather quite small, distinctive peptides (11aa), a feature that already limits the possibility of crossed reactions in the antisera. Our negative controls (see new Supplementary Figure 2 at the bottom of this letter) and the immune-labelling pattern of target proteins in the spicules also indicate that our antibodies are specific, despite the disparity of labelling intensity on the gels. Intensity is not always related to specificity. Please, read this letter to the end before making further objections in this regard.

The reviewer also indicates that “the authors consider higher molecular weight bands to be oligomers, but this is speculation and it is unclear why they would have oligomers if reducing agents were added to disrupt disulphide bonds.” Please, note that when the reviewer indicates that this is speculation, it is being ignored the fact that such a pattern has been reported in the literature for many other proteins. There are not few reports in the literature indicating non-covalent dimerization of proteins in gels (e.g., Gilda et al., 2015), also covalent binding other than disulfide (e.g., Kobayashi et al., 2019). In fact, oxidation of proteins reduced with dithiothreitol or 2-mercaptoethanol occurs during SDS-PAGE, resulting in appearance of dimers, trimers and oligomers on the gels, in addition to the monomers (Achilli et al., 2018). There is also a forum where students and scientists share discussion about how they could deal with this problem in their gels, which can derive from samples insufficiently reduced but also from other processes in properly reduced samples (e.g., https://www.researchgate.net/post/Dimers_at_western_blot_but_not_for_all_samples).

Note also that, in our case, the “polymeric bands” do not only correspond well to the expected molecular weights, but also appear labeled by their respective specific antibodies as would be expected. Besides, note that we only interpret these bands as “PUTATIVE” dimers and/or trimers. Therefore, we are not speculating so much as interpreting relatively obvious patterns.

In response to this comment, we have modified the text of legend of Supplementary Fig. 1 as “Note the appearance of faint bands of putative dimers and trimers despite the use of reducing agents, bands that are specifically recognized by their respective antibodies.”

- 2) the Western Blots for *Vazella* hexaxilin and perisilin are very unconvincing. Indeed, there are multiple bands detected, and the authors tell us what they think those bands could be, but there is no way of knowing with any certainty.

AUTHOR RESPONSE: We disagree that we are providing unconvincing evidence of our interpretations. We have already addressed this in the previous comment, but we explain a concrete example here. For instance, in the case of *Vazella* hexaxilin in Figure S1e and f, there are two bands, one corresponding to the molecular weight of hexaxilin and the other corresponding to twice that MW and both bands are recognized by the antibody against hexaxilin. We would not call this “unconvincing evidence”. Then, we also have the staining pattern of hexaxilin by anti-hexaxilin in the spicules of *Vazella* in main Fig. 2, along with the pre-immune control of it in new Supplementary Figure 2 at the bottom of this letter. Honestly, we admit that some of our gels could be of better quality, but to conclude that our evidence is poor or “unconvincing” is quite exaggerated, in our opinion.

- 3) The only other validation measure presented are Elisa results from Sigma after antibody production. These only show that the rabbits did produce antibodies against the injected immunogens but not that the antibodies are specific and recognize the endogenous, folded protein in the sponges. It remains unclear why the authors wouldn't affinity purify their antibodies against the injected antigen, and despite clear and interpretable immunostaining patterns there remains uncertainty about whether these antibodies are specific to the intended endogenous target.

AUTHOR RESPONSE: We fully understand the general benefits to purifying antisera, but, contrary to the reviewer’s assertion, antisera are also used successfully (and with high specificity in response) in many applications. Please, see the results below of a study comparing the performance of a rabbit antiserum (Fig 1 below) and an affinity purified polyclonal antibody (Fig. 2 below) against human protein GPR30 (Manjegowda et al., 2016). There are no differences in the outcome. In fact, the study concludes textually that *“The affinity purified antibody obtained from the antiserum of rabbit B showed similar reactivity to that of the antiserum”*. This is the reason why, as the reviewer acknowledges, *“clear and interpretable immunostaining patterns”* also resulted with our antisera. Sorry, but we must insist: our antibodies have no specificity problems. Although we admit that some technical aspects in some particular steps of the global process could have been improved, the overall outcome of our immunostaining is valid.

Fig. 1. Quality assessment of antiserum against N-terminus of GPR30. Protein lysates prepared from a panel of breast cancer cell lines were fractionated by 10% SDS-PAGE under denaturing conditions and transferred to nitrocellulose membrane. Membranes were subjected to western blotting analysis followed by chemiluminescence detection. The primary antibodies for each of the above panels are- A. 1 in 10,000 dilution of antiserum from Rabbit B; B. 1 in 10,000 dilution of pre-immune serum from Rabbit B; C. 1 in 5000 dilution of commercial anti- β -actin antibody. Image and legend from Manjegowda et al. 2016.

Fig. 2. Detection of GPR30 in total protein by affinity purified antibody. Proteins were fractionated by 10% SDS-PAGE under denaturing conditions and transferred to nitrocellulose membranes. Membranes were subjected to western blotting analysis followed by chemiluminescence detection. The anti- β -actin antibody

was used in a dilution of 1:5000 and the affinity purified primary antibody was used in a dilution of 1:15,000. Image and legend from Manjgowda et al. 2016.

We wonder why additional tests would still be necessary to characterize, for instance, hexaxilin, when we have already shown that the protein can be extracted from within the silica, that it is overexpressed in abundance of silicate, that its antibodies stain western blot gels reasonably well and also stain quite specifically the axial filament of two different species. We cannot understand why the reviewer is not convinced. Most of the concerns raised would apply to approaches using whole proteins as immunogens. However, this is not the case. We have used tiny, highly distinctive peptides and the immunostaining patterns obtained are all coherent and consistent (see also our response to your last comment on controls for our immunostaining). Please, take into consideration the general agreement among the various constituents of this study and the global, coherent message that they compose.

Regarding immunostaining patterns and faint peripheral staining of hexaxilin, the authors suggest that this is due to antibody getting trapped in spicule fractures and being inadequately washed out. This is possible, but there is no evidence to support this interpretation. In fact, if this interpretation were correct, then the pre-immune sera control and/or secondary-only control should give the same pattern, but this is not shown. I think it is at least possible that hexaxilin is present at lower abundance in peripheral areas of the spicule too. I stand by my opinion that this study is exciting and important. I've highlighted the weaknesses that I believe remain, but these should not preclude publication of the manuscript.

AUTHOR RESPONSE: Admittedly, we did not include images of the controls with pre-immune serum in the manuscript. Images of such routine controls are not very appealing to readers. We merely mentioned that the controls worked ok. However, since we did those tests for internal use and the reviewer is requesting additional evidence of such controls to accept our interpretations on the role and location of hexaxilin, we provide the images here (see plate below). As you will see, our SEM images of the spicules and the staining pattern resulting from using pre-immune serum as a control show that part of the fluorochrome is capillary adsorbed under the outermost layer of silica (which is quite loosely packed) during the incubation process. This fluorochrome cannot be completely removed during the subsequent blocking and rinsing steps. Therefore, our interpretation was correct: anti-hexaxilin is specific only against the axial filament, as already indicated in the Main Text of the manuscript and its Figure 2. Contrary to this reviewer's suggestion, hexaxilin does not occur in the axial filament and in the outermost peripheral layer of the silica, but only in the axial filament. Please also note that, according to our current understanding of silicification cytology, such a dual spatial distribution for hexaxilin would be hard to be explained.

The new figure presented below in response to the reviewer's comment will constitute the new Supplementary Figure 2 of the revised manuscript (R2).

Supplementary Fig. 2. Effects of a loose outermost silica layer on fluorochrome distribution. (a-b) Scanning electron microscopy views of spicules of *Vazella pourtalesii* showing that the outermost layer of silica (ol) consists of sublayers that are loosely attached to each other and relative to the internal silica core (il) of the spicule. Such a feature is known to be more pronounced in *V. pourtalesii* and other rossellids than in another hexactinellid groups (Maldonado et al., 2019, 2022). **(c-d)** Spicule of *V. pourtalesii* incubated with pre-immune serum against hexaxilin. Note in the bright field image (c) the occurrence of portions with broken silica (white arrows) and loose peripheral silica (yellow arrows), both of which adsorb the fluorochrome by capillarity, which subsequently cannot be completely removed even after several rinsing steps, causing a very weak staining that might lead to erroneous interpretations about the specificity of the antibodies for the axial filament. **(e-f)** Spicule of *Euplecetella curvistellata* incubated with pre-immune serum against hexaxilin. Note

that the adsorption effect by the peripheral silica is less pronounced, because the layers are less loose than in *V. pourtalesii*. These controls with pre-immune serum indicate that, as shown in Fig. 2 of main text, the designed anti-hexaxilin antibodies label with specificity only the protein in the axial filament but no protein constituents of the peripheral silica.

REFERENCES

- Achilli, C., Ciana, A., and Minetti, G. (2018). Oxidation of cysteine-rich proteins during gel electrophoresis. *J. Biol. Methods* 5, e104. doi:10.14440/jbm.2018.275.
- Charette, S. J., Lambert, H., Nadeau, P. J., and Landry, J. (2010). Protein quantification by chemiluminescent Western blotting: Elimination of the antibody factor by dilution series and calibration curve. *J. Immunol. Methods* 353, 148–150. doi:10.1016/j.jim.2009.12.007.
- Gilda, J. E., Ghosh, R., Cheah, J. X., West, T. M., Bodine, S. C., and Gomes, A. V. (2015). Western blotting inaccuracies with unverified antibodies: Need for a Western Blotting Minimal Reporting Standard (WBMRS). *PLoS One* 10. doi:10.1371/journal.pone.0135392.
- Gunning, P. W., and Hardeman, E. C. (2018). Fundamental differences. *Elife* 7. doi:10.7554/ELIFE.34477.
- Kobayashi, M., Muramatsu, K., Haruyama, T., Uesugi, H., Kikuchi, A., Konno, H., et al. (2019). Polymerization of Oxidized DJ-1 via Noncovalent and Covalent Binding: Significance of Disulfide Bond Formation. *ACS Omega* 4, 9603–9614. doi:10.1021/ACSOMEGA.9B00324/SUPPL_FILE/AO9B00324_SI_004.MOV.
- Maldonado, M., López-Acosta, M., Abalde, S., Martos, I., Ehrlich, H., and Leynaert, A. (2022). On the dissolution of sponge silica: Assessing variability and biogeochemical implications. *Front. Mar. Sci.* 9, 2487. doi:10.3389/FMARS.2022.1005068.
- Maldonado, M., López-Acosta, M., Sitjà, C., García-Puig, M., Galobart, C., Ercilla, G., et al. (2019). Sponge skeletons as an important sink of silicon in the global oceans. *Nat. Geosci.* 12, 815–822. doi:10.1038/s41561-019-0430-7.
- Manjegowda, M. C., Gupta, P. S., and Limaye, A. M. (2016). Validation data of a rabbit antiserum and affinity purified polyclonal antibody against the N-terminus of human GPR30. *Data Br.* 7, 1015–1020. doi:10.1016/j.dib.2016.03.054.
- Mohri, K., Nakatsukasa, M., Masuda, Y., Agata, K., and Funayama, N. (2008). Toward understanding the morphogenesis of siliceous spicules in freshwater sponge: Differential mRNA expression of spicule-type-specific silicatein genes in *Ephydatia fluviatilis*. *Dev. Dyn.* 237, 3024–3039. doi:10.1002/dvdy.21708.
- Müller, W. E. G., Schloßmacher, U., Eckert, C., Krasko, A., Boreiko, A., Ushijima, H., et al. (2007). Analysis of the axial filament in spicules of the demosponge *Geodia cydonium*: Different silicatein composition in microscleres (asters) and megascleres (oxeas and triaenes). *Eur. J. Cell Biol.* 86, 473–487. doi:10.1016/j.ejcb.2007.06.002.
- Perrin, B. J., and Ervasti, J. M. (2010). The actin gene family: Function follows isoform. *Cytoskeleton* 67, 630–634. doi:10.1002/CM.20475.

REVIEWERS' COMMENTS

Reviewer #2 (Remarks to the Author):

I have read the authors' rebuttal to my remaining criticisms of their manuscript. As I indicated in my last review - I was making these points for the authors to consider, but there were no remaining issues that should preclude publication. My position hasn't changed. My recommendation is to publish this manuscript without further changes.